# Entropy testing and its application to testing Bayesian networks

**Clément L. Canonne**
University of Sydney
clement.canonne@sydney.edu.au

**Joy Qiping Yang**
University of Sydney
qyan6238@uni.sydney.edu.au

## Abstract

This paper studies the problem of *entropy identity testing*: given sample access to a distribution $p$ and a fully described distribution $q$ (both discrete distributions over a domain of size $k$), and the promise that either $p = q$ or $|H(p) - H(q)| \geqslant \varepsilon$, where $H(\cdot)$ denotes the Shannon entropy, a tester needs to distinguish between the two cases with high probability. We establish a near-optimal sample complexity bound of $\tilde{\Theta}(\sqrt{k}/\varepsilon + 1/\varepsilon^2)$ for this problem, and show how to apply it to the problem of identity testing for in-degree-$d$ $n$-dimensional Bayesian networks, obtaining an upper bound of $\tilde{O}(2^{d/2}n^{3/2}/\varepsilon^2 + n^2/\varepsilon^4)$. This improves on the sample complexity bound of $\tilde{O}(2^{d/2}n^2/\varepsilon^4)$ from [CDKS20], which required an additional assumption on the structure of the (unknown) Bayesian network.

## 1 Introduction

Entropy is a fundamental information theory notion, which quantifies the amount of "uncertainty" a given random variable carries. Since its introduction by Shannon, this notion has found myriads of applications, and is central – among others – to compression and coding, probability, electrical engineering, and learning theory.

As a result, the task of *estimating* the Shannon entropy of a discrete random variable (or, equivalently, its probability distribution) from samples has naturally emerged, starting (in Computer Science) with the work of [BDKR02] which considered *multiplicative* approximations. *Additive* approximation of the entropy (within $\pm\varepsilon$) was then considered in a series of papers [VV11a, VV11b, VV13, HJW15a, ADOS17], culminating with the work of [WY16], which establishes the optimal sample complexity, $\Theta\left(\frac{k}{\varepsilon \log k} + \frac{\log^2 k}{\varepsilon^2}\right)$, where $k \gg 1$ is the domain size.

While the resulting sample complexity is *sublinear* in the domain size $k$, it is only so by a mere logarithmic factor. In some settings, paying this near-linear dependence in the amount of data necessary is impractical, typically in the large-domain regime (e.g., for high-dimensional data, where $k$ is exponential in the dimension); moreover, it may even be *unnecessary*. Specifically, one may not be concerned so much about the (approximate) value of the entropy of a distribution, but rather about whether it is above a threshold, or differs from that of a given purported model.

It is this latter task we introduce and consider in our work, which can be seen as a variant of the standard *identity testing* question from distribution testing: given a reference known hypothesis distribution $q$ over a domain of size $k$, and i.i.d. samples from an unknown distribution $p$, what is the sample complexity of testing whether $p$ is equal to $q$, or their entropies differ significantly? And, crucially, *is this testing task more sample-efficient than that of estimating $H(p)$?*

38th Conference on Neural Information Processing Systems (NeurIPS 2024).

> **Entropy Identity testing:** Given a reference distribution $q$, parameter $\varepsilon > 0$, and samples from an unknown $p$, what is the cost of deciding (with high probability) whether $p = q$ vs. $|H(p) - H(q)| > \varepsilon$, with correct probability at least $2/3$?

Note that in the case where $q$ is the uniform distribution over the domain, this task is equivalent to distinguishing between $H(p) = \log k$ and $H(p) < \log k - \varepsilon$.

Our main contribution is to show that the testing question can indeed be performed much more efficiently than the estimation one, at least for most parameter regimes. Specifically, we establish the following theorem:

**Theorem 1.1.** *The sample complexity of entropy identity testing is $O(\sqrt{k \log(k/\varepsilon)}/\varepsilon + \log^2(k)/\varepsilon^2)$. Moreover, this is nearly tight: $\Omega(\sqrt{k}/\varepsilon + \log^2 k/\varepsilon^2)$ samples are necessary in the worst case.*

Interestingly, this differs both from the *estimation* task (which, as discussed before, has a near-linear dependence on the domain size $k$) but also from identity testing *in total variation distance*, which has sample complexity $\Theta(\sqrt{k}/\varepsilon^2)$ (see Section 1.1).

**Application: Identity testing for Bayesian networks.** As an application of Theorem 1.1, we derive an efficient algorithm for identity testing (in total variation distance) for maximum in-degree $d$ Bayesian networks (shorten as degree-$d$ Bayes net in the remaining of the paper):[1]

**Theorem 1.2** (Informal; see Theorem 3.1). *There is an algorithm which, given sample access to a degree-$d$ Bayes net $p$ and the full description of a reference degree-$d$ Bayes net $q$ (both over $\{0,1\}^n$), takes $\tilde{O}\left(\frac{2^{d/2}n^{3/2}}{\varepsilon^2} + \frac{n^2}{\varepsilon^4}\right)$ samples from $p$, and distinguishes between $p = q$ and $d_{\mathrm{TV}}(p,q) \geq \varepsilon$.*

Prior to this, the best known sample complexity upper bound for this task [CDKS20] was quadratically worse in both $n$ and $\varepsilon$, and further required an assumption on the underlying graph structure of both $p$ and $q$. We emphasize that (1) our result improves on the sample complexity of the learning baseline for $d \gg \log(n/\varepsilon)$, and on its computational efficiency; and (2) compared to the previous testing results, removes strong structural assumptions which considerably limited their applicability. We elaborate on this in the next section.

## 1.1 Related work

As previously discussed, entropy estimation has received a considerable amount of interest from computer scientists, information theorists and statisticians [BDKR02, Pan04, HJW15a, WY16]. Entropy is also a key example of *symmetric property* (invariant to relabeling of the domain) [VV11a, VV11b, VV13, ADOS17], and has been considered in other settings as well, e.g., the quantum case [GHS21, AISW20] and the memory-limited setting [ABIS19, AMNW22]. Estimation of some generalizations of Shannon entropy, such as the family of Rényi entropies, also have been studied [AOST17].

Over the years, sample complexity of identity testing for discrete distribution has been intensively studied and essentially settled [Pan08, BFF+01, VV17]. In high dimensions, however, the square root dependence of the sample complexity on the domain size means that most identity testing tasks of interest require sample complexity exponential in the dimension. Moreover, this curse of dimensionality extends to a large range of distribution testing problems [BCY22, Theorem B.1]. As such, many turn to the study of testing distributions under additional natural structural assumptions, such as graphical models: [BGKV21] look at identity testing for product distributions (degree-0 Bayes nets) and give the optimal bound of $\Theta(\sqrt{n|\Sigma|}/\varepsilon^2)$, where $|\Sigma|$ is the alphabet size of each variable (rather than binary alphabet studied in our paper). [DDK19, KDDC23] study testing Ising models, obtaining sample complexity bounds that are $\mathrm{poly}(n/\varepsilon)$; [DP16], [CDKS20] give tight results to identity testing and closeness testing for a variety of constant in-degree Bayes nets, which also gives polynomial sample complexity bounds.

However, the testing algorithms provided in [CDKS20] and [DP16] are not fully satisfactory, as they require some strong assumptions on Bayes nets. Specifically, [CDKS20, Theorem 21] assumes that

---

[1]Our algorithm actually provides a stronger guarantee, with respect to Hellinger distance, which implies the TV result as $d_{\mathrm{TV}}(p,q) \leq \sqrt{2}\, d_{\mathrm{H}}(p,q)$ for any two distributions $p, q$.

the topological ordering of the two Bayes nets are the same, and shows that under this assumption $O(2^{d/2}n^2/\varepsilon^4)$ samples are sufficient.[2] [CDKS20, Theorem 17] makes the further stringent restriction that the reference Bayes net has to be *balanced*, i.e., that the conditional probabilities are all bounded away from 0 and 1; moreover, it also requires every parental configuration to be bounded from 0, and that the structure of the unknown Bayes net be a subset of that of the reference one. The result of [DP16, Theorem 4.2] combined with the Hellinger tester from [DKW18, Theorem 1] implies that, under the assumption that $p$ and $q$ share the same factorization structure (i.e., their associated DAGs are the same or one is a subgraph of the other), then this problem is solvable in $\tilde{O}\left(2^{d/2}n/\varepsilon^2\right)$ samples. While this latter sample complexity is near-optimal (in some regime[3]), in view of the $\Omega\left(2^{d/2}n/\varepsilon^2\right)$ lower bound obtained in [BCY22, Theorem 4.1], the factorization structure requirement considerably limits the applicability of the algorithm.

One can also compare our result to the *learning* results on Bayesian networks, as any learning algorithms enables testing as well (the "testing-by-learning" baseline). It is known [CDKS20]) that learning degree-$d$ Bayes nets can be done with $\tilde{O}(2^d n/\varepsilon^2)$ samples, without any structural assumptions. Our testing result improves on this sample complexity as long as $n^2/\varepsilon^4 \ll 2^d n/\varepsilon^2$ and $2^{d/2}n^{3/2} \ll 2^d n$, i.e., for $d \gg \log(n/\varepsilon)$; moreover, it is worth noting that the known learning algorithms are computationally inefficient (running in time $n^{O(dn)}$ via an enumeration of all possible underlying graph structures [CDKS20, BGMV20]), and this is believed to be inherent [CHM04]. In contrast, our algorithm runs in time $\mathrm{poly}(n^d, 1/\varepsilon)$.

## 1.2 Techniques overview

**Testing in entropy.** A first idea is to use the conversion between total variation (TV) distance and entropy difference to reduce this problem to identity testing in TV: When $d_{\mathrm{TV}}(p,q) \leqslant 1/2$, then $|H(p) - H(q)| \leqslant d_{\mathrm{TV}}(p,q) \log \frac{k}{d_{\mathrm{TV}}(p,q)}$ [CK11, Lemma 2.7].[4] This gives an upper bound of $O(\frac{\sqrt{k}\log^2(k/\varepsilon)}{\varepsilon^2})$, which is already better than the sample complexity of estimation: $O\left(\frac{k}{\varepsilon \log k} + \frac{\log^2 k}{\varepsilon^2}\right)$ for the parameter $k$. However, it is not clear whether the quadratic dependence on $\varepsilon$ is necessary: indeed, the "hard instances" for TV testing (the Paninski construction [Pan08]), small perturbations around the uniform distribution which have TV distance $\varepsilon$ from uniform, actually only have entropy $\log k - \Theta(\varepsilon^2)$. The $\Omega(\sqrt{k}/\varepsilon^2)$ uniformity testing lower bound from these hard instances thus only implies an $\Omega(\sqrt{k}/\varepsilon)$ entropy identity testing lower bound!

A next natural idea is to strengthen the lower bound. However, it then becomes clear that the Paninski [Pan08] construction cannot be improved: as just mentioned, when its TV distance to the uniform distribution is around $\Theta(\sqrt{\varepsilon})$ its entropy difference to it is only $\Theta(\varepsilon)$ (giving an $\Omega(\sqrt{k}/\varepsilon)$ lower bound). Moreover, this is not a coincidence: when the reference distribution $q$ is uniform, we are able to get a matching upper bound using [DKW18, Algorithm 1], upon noticing that

$$H(p) = \log k - d_{\mathrm{KL}}(p\|u_k), \tag{1}$$

which implies $d_{\mathrm{KL}}(p\|u_k) = \log k - H(p) \geqslant \varepsilon$, where $u_k$ is the uniform distribution on $[k]$ and $d_{\mathrm{KL}}$ denotes the Kullback–Leibler divergence. Interestingly, a completely different hard instance, against a very much non-uniform reference distribution, does yield the second term of our lower bound, $\Omega(\log^2 k/\varepsilon^2)$.

Inspired by these two different lower bounds, we can generalize (1) by defining $\mathcal{A}$ as the set of "not too small probability elements under $q$", and then observing (looking ahead, using the inequality (7)) that

$$|H(p_{\mathcal{A}}) - H(q_{\mathcal{A}})| \leqslant |d_{\mathrm{KL}}(p_{\mathcal{A}}\|q_{\mathcal{A}})| + \left|\sum_{i \in \mathcal{A}}(p_i - q_i)\log\frac{1}{q_i}\right| \tag{2}$$

where $H(p_{\mathcal{A}})$ is the "entropy" of the sub-distribution restricted to the set $\mathcal{A}$. In particular, this hints that one could solve the general problem by testing if either of the two terms on the right-hand-side

---

[2]While the sample complexity of the algorithm is not explicitly stated in their proof, inspection of their argument yields this bound.

[3]The lower bound [BCY22, Theorem 4.1] only holds under the sparse regime: $d \ll \log n$.

[4]All logarithms in the paper are natural ($e$ as base).

is large. The name of the game now is to (i) choose the threshold for $\mathcal{A}$ (i.e., what does it mean for an element to have "not too small probability under $q$"), and (ii) have algorithms to test whether these two quantities are noticeably large.

Let us focus on how to test the first term of (2). If $\min_i q_i \geqslant \Omega\left(\frac{\varepsilon}{k}\right)$, we can adapt and use an algorithm of [DKW18] to efficiently test $d_{\mathrm{KL}}(p\|q) \geqslant \varepsilon$ vs. $p = q$. In addition, if $\log(1/q_i)$ is bounded, then in fact, estimating the second term to $O(\varepsilon)$ is possible as well. Thus it is natural to wonder if we can afford to neglect the region where $q_i \leqslant \frac{\varepsilon}{k}$. Indeed, the impact on entropy is at most $O(\tau \log(k/\tau))$ if we are to remove regions with at most $O(\tau)$ as mass. Thus, by adjusting the appropriate threshold, we can still detect difference in entropy even if we only test on elements with greater than $\tau/k$ masses, where $\tau = \frac{\varepsilon}{\log(k/\varepsilon)}$.

The problem then becomes to check if $p$ puts more than $100\tau$ mass in $\bar{\mathcal{A}} = \{i \in [k] : q_i < \tau/k\}$, which costs $O(1/\tau) = O(\log(k/\varepsilon)/\varepsilon)$ samples. If it does, then it cannot be the case that $p = q$; we can reject. After this stage, both $p(\bar{\mathcal{A}}), q(\bar{\mathcal{A}}) \leqslant O(\tau)$. To move forward, we need to check the influence on entropy: $H(p)$ and $H(q)$. By Jensen's inequality and monotonicity of $f(x) = x \log \frac{1}{x}$ when $x < \frac{1}{e}$, we have

$$\sum_{i \in \bar{\mathcal{A}}} p_i \log \frac{1}{p_i} \leqslant p(\bar{\mathcal{A}}) \log \frac{k}{p(\bar{\mathcal{A}})} \leqslant \tau \log \frac{k}{\tau}.$$

Therefore, the impact on entropy will be at most $O\left(\tau \log \frac{k}{\tau}\right)$. Setting $\tau = \frac{\varepsilon}{\log(k/\varepsilon)}$, this becomes $O(\varepsilon)$, which gives us the room to check if $|H(p_{\mathcal{A}}) - H(p_{\mathcal{A}})| \geqslant 100\varepsilon$ or $p_{\mathcal{A}} = q_{\mathcal{A}}$.

**Testing Bayesian networks.** Similar to [DP16, Theorem 4.2],[5] the identity testing algorithm is straight-forward: check all every $i \in [n]$, if $p_{X_i, \Pi_i^G} = q_{X_i, \Pi_i^G}$ or is one of them is far apart, where $q$ is Markov with respect to $G$ ($q$ factorizes according the DAG $G$). The main technical part is to show that the distance is "subaddititve" when $p$ and $q$ share no common structure, but are close to sharing a common factorization structure (this can be thought of as a relaxtion of [DP16, Theorem 4.2]; refer to Lemma 3.3 for details). As a consequence of "subadditivity", if $p$ and $q$ are far in distributional distance (differed from [DP16], our work opted to test in KL divergence restricted on subsets with large enough density), then it would imply that one of the local distance between $p$ and $q$ is sufficiently large. This allows us to reduce from global testing to local testing.

Another key aspect is checking whether $p$ and $q$ are close to sharing common structure. More specifically, whether $d_{\mathrm{KL}}(p\|p_G)$ is small, where $p_G$ is the projection of $p$ unto $q$'s DAG $G$. Here, we establish a connection between entropy closeness and structure closeness. In particular, we show that if every local entropy (involving subsets of size $d + 1$, where $d$ is the bound on maximum in-degree) between $p$ and $q$ is close, then this means that they must approximately share the same structure (see Lemma 3.4). The intuition behind the connection is that if all tests pass, then we can conclude that $p$ and $q$ are close in local entropy and thereafter, we can utilize entropy of $q$ to learn the graphical structure [KCG+23] of $p$ (which uses no additional samples).

At a high level, our algorithm first check if $p$ and $q$ roughly share the same structure via a proxy check of local entropy tests. If all local entropy tests pass, then we can show that there exists $i \in [n]$ such that local KL restricted on subset with large enough mass is greater than $\Omega\left(\frac{\varepsilon^2}{n}\right)$. A subsequent identity test with $\chi^2$-test [DKW18] suffice.

**Preliminaries and notation.** The (Shannon) entropy $H$ of a discrete distribution $p$ supported on $[k]$ is given by:

$$H(p) = -\sum_{i \in [k]} p_i \log p_i.$$

The conditional entropy $H(p_X \mid p_Y)$ for $X$ supported on $\mathcal{X}$, and $Y$ on $\mathcal{Y}$, defined by the joint distribution $p_{X,Y}$, can be written as

$$H(p_X \mid p_Y) = -\sum_{x \in \mathcal{X}, y \in \mathcal{Y}} p(x, y) \log \frac{p(x, y)}{p(y)} = H(p_{X,Y}) - H(p_Y). \tag{3}$$

---

[5]We note that what they refer to as "identity testing" is different from ours (and the standard) use of the term: in their setting, the reference distribution is replaced with sample access to the distribution (this is commonly referred to as "closeness testing").

We adopt the entropy notation for a sub-probability vector $H(q_{\mathcal{A}}) = \sum_{i \in \mathcal{A}} q_i \log \frac{1}{q_i}$. Throughout this paper, we will use $e$ as base of the log and of the entropy. We will use $\leftarrow$ for variable assignment. We adopt the standard $O(\cdot)$, $\Omega(\cdot)$ and $\Theta(\cdot)$ asymptotic notation and use $\tilde{\cdot}$ to hide any polylogarithmic factors in the argument. We will use various metrics or divergences on probability distributions: Kullback–Leibler ($d_{\mathrm{KL}}$), Hellinger ($d_{\mathrm{H}}$), chi-squared ($d_{\chi^2}$), and total variation ($d_{\mathrm{TV}}$). We denote $p_{\mathcal{A}}$ as restricting $p$ onto the elements in $\mathcal{A}$, and we denote distributional distances restricting on $\mathcal{A}$ as follows: $d_{\mathrm{KL}}(p_{\mathcal{A}}, q_{\mathcal{A}}) = \sum_{i \in \mathcal{A}} p_i \log \frac{p_i}{q_i}$. $d_{\mathrm{H}}(p_{\mathcal{A}}, q_{\mathcal{A}}) = \frac{1}{\sqrt{2}} \sqrt{\sum_{i \in \mathcal{A}} \left( \sqrt{p_i} - \sqrt{q_i} \right)^2}$. For a set $\mathcal{A}$, we write $p(\mathcal{A}) = \sum_{i \in \mathcal{A}} p_i$. We also have the following inequality [DKW18, Proposition 1]:

$$d_{\mathrm{TV}}(p_{\mathcal{A}}, q_{\mathcal{A}}) \leqslant \sqrt{2}\, d_{\mathrm{H}}(p_{\mathcal{A}}, q_{\mathcal{A}}) \leqslant \sqrt{\sum_{i \in \mathcal{A}}(q_i - p_i) + d_{\mathrm{KL}}(p_{\mathcal{A}}, q_{\mathcal{A}})} \leqslant \sqrt{d_{\chi^2}(p_{\mathcal{A}}, q_{\mathcal{A}})}. \quad (4)$$

A distribution $p$ supported over the hypercube $\{0, 1\}^n$ is a Bayesian network if its probability mass function satisfies the factorization associated with $G$, a directed acyclic graph (DAG):

$$p(x_1, \cdots, x_n) = \prod_{i=1}^{n} p(x_i | \Pi_i), \quad (5)$$

and $\Pi_i$ is the set of parents of $X_i$ in $G$; and we say that $p$ is Markov with respect to DAG $G$. In section 3, slightly abusing notation, we use $p_G$ to denote a projection of a Bayes net $p$ to a DAG $G$ (which it may or may not be Markov with respect to; see Definition 3.2). We work in the Poissonized setting (see, e.g., [Can22, Appendix C]) – instead of drawing $N$ samples directly from $p$, we draw $Y \sim \mathrm{Poi}(N)$ samples from $p$, where $\mathrm{Poi}(N)$ denotes the random variable distributed as the Poisson distribution with parameter $N$. The Poissonized and usual sampling settings are equivalent for constant probability of failure, up to a (small) multiplicative factor in the sample complexity.

## 2 Near-optimal entropy testing

We prove Theorem 1.1, establishing the sample complexity upper and lower bounds separately.

### 2.1 An $O\big(\frac{\sqrt{k \log(k/\varepsilon)}}{\varepsilon} + \frac{\log^2(k)}{\varepsilon^2}\big)$ upper bound

We will prove the following theorem:

**Theorem 2.1.** *There is an algorithm (Algorithm 1) which, given $n$ samples from a discrete distribution $p$, the full description of a reference distribution $q$, both over $[k]$, and parameter $\varepsilon > 0$, distinguishes between $p = q$ and $|H(p) - H(q)| \geqslant \varepsilon$ with probability at least $2/3$, as long as*

$$n \geq c_1 \left( \frac{\sqrt{k \log(k/\varepsilon)}}{\varepsilon} + \frac{\log^2(k)}{\varepsilon^2} \right)$$

*and $c_2 \varepsilon \leqslant k$, for some absolute constants $c_1, c_2 > 0$. Moreover, the algorithm runs in time linear in the number of samples $n$ and the domain size $k$.*

The proof will rely on the two following claims and Lemma 2.4, which is a straightforward extension of [DKW18, Lemma 2]. Their proofs are deferred to Appendix B. Throughout, we let $\tau := \frac{\varepsilon}{16 \log(k/\varepsilon)}$, and $\mathcal{A} := \big\{ i \in [k] \mid q_i \geqslant \frac{\tau}{k} \big\}$, as in Algorithm 1.

**Claim 2.2.** *Let $\mathcal{A}$ be any set such that $p(\bar{\mathcal{A}}) < \varepsilon/2$. Then, if $|H(p_{\mathcal{A}}) - H(q_{\mathcal{A}})| \geqslant \varepsilon$, we must have (i) $d_{\mathrm{KL}}(p_{\mathcal{A}} \| q_{\mathcal{A}}) \geqslant \frac{\varepsilon}{2}$ or (ii) $|\sum_{i \in \mathcal{A}}(p_i - q_i) \log(\frac{1}{q_i})| \geqslant \frac{\varepsilon}{2}$.*

**Claim 2.3.** *Let $\hat{p}$ be the empirical estimator for an unknown discrete distribution $p$ supported on $[k]$, based on $\mathrm{Poi}(m)$ samples, where $m = \Theta\left( \frac{\log^2(k)}{\varepsilon^2} \right)$; assume that $d_{\chi^2}(p_{\mathcal{A}}, q_{\mathcal{A}}) \leqslant \varepsilon/8$ and $p(\bar{\mathcal{A}}) + q(\bar{\mathcal{A}}) \leqslant 4\tau = \frac{1}{4} \frac{\varepsilon}{\log(k/\varepsilon)}$,[6] then*

$$\Pr\left[ \left| \sum_{i \in \mathcal{A}}(p_i - \hat{p}_i) \log \frac{1}{q_i} \right| \geqslant \frac{1}{8}\varepsilon \right] \leqslant \frac{1}{100}.$$

---

[6]One can remove the assumption that $p(\bar{\mathcal{A}}) + q(\bar{\mathcal{A}}) \leqslant 4\tau$, at the cost of a slightly worse constant.

---

**Algorithm 1** Entropy identity testing

---

**Require:** Sample access to $p$ and full description of $q$, both over $[k]$; accuracy parameter $\varepsilon$.

1: Set $\tau := \frac{\varepsilon}{16\log(k/\varepsilon)}$, and $\mathcal{A} := \left\{ i \in [k] \mid q_i \geqslant \frac{\tau}{k} \right\}$.

2: Take $m_1 = 48/\tau$ samples from $p$ and compute the empirical $\hat{p}'$.

3: Compute $Z_1 = \hat{p}'(\bar{\mathcal{A}})$.

4: **if** $Z_1 \geqslant 2\tau$ **then return** reject $\qquad\qquad\qquad\qquad\qquad\qquad\qquad$ ▷ Early rejection.

$\qquad\qquad\qquad\qquad\qquad\qquad$ ▷ $N_i$: the empirical count among samples of the $i$-th element.

5: Let $m_2 = 65536 \left( \frac{\sqrt{k \cdot \log(k/\varepsilon)}}{\varepsilon} \right)$. Draw $\mathrm{Poi}(m_2)$ samples from $p$ and compute

$$Z_2 = \sum_{i \in \mathcal{A}} \frac{(N_i - Nq_i)^2 - N_i}{Nq_i}.$$

6: **if** $Z_2 \geqslant \frac{1}{16} m_2 \varepsilon$ **then return** reject

7: Let $m_3 = 140800 \left( \frac{\log^2(k)}{\varepsilon^2} \right)$

8: Draw $\mathrm{Poi}(m_3)$ samples from $p$, compute the empirical $\hat{p}$; let $Z_3 \leftarrow \left| \sum_i (\hat{p}_i - q_i) \log \left( \frac{1}{q_i} \right) \right|$.

9: **if** $Z_3 \geqslant \frac{1}{8} \varepsilon$ **then return** reject

10: **return** accept

---

**Lemma 2.4.** *Let* $\mathcal{A} := \{ i \in [k] \mid q_i \geqslant \alpha \}$. *Let* $m_2 \geqslant 16384 \max \left\{ \sqrt{\frac{1}{\alpha\varepsilon}}, \frac{\sqrt{k}}{\varepsilon} \right\}$ *be the number of samples used to compute* $Z_2$. *Then* $\mathbb{E}[Z_2] = m_2 d_{\chi^2}(p_{\mathcal{A}}, q_{\mathcal{A}})$. *Moreover, if* $d_{\chi^2}(p_{\mathcal{A}}, q_{\mathcal{A}}) \leqslant \frac{\varepsilon}{2}$, *then* $\mathrm{Var}[Z_2] \leqslant (\frac{1}{32} m_2 \varepsilon)^2$. *If* $d_{\chi^2}(p_{\mathcal{A}}, q_{\mathcal{A}}) \geqslant \varepsilon$, *then* $\mathrm{Var}[Z_2] \leqslant O(\mathbb{E}[Z_2]^2)$.

*Proof of Theorem 2.1.* We prove the statement by analyzing Algorithm 1. First, note that excluding the set of $\bar{\mathcal{A}}$ (elements with small mass), can change the value of $H(q)$ by at most $\varepsilon/8$: indeed, by Jensen's inequality ($f(x) = \log x$ is concave) and $x \log \frac{1}{x}$ being monotonically increasing in $(0, 1/e)$,

$$H(q_{\bar{\mathcal{A}}}) = \sum_{i \in \bar{\mathcal{A}}} q_i \log \frac{1}{q_i} \leqslant q(\bar{\mathcal{A}}) \log \frac{|\bar{\mathcal{A}}|}{q(\bar{\mathcal{A}})} \leqslant \tau \log \frac{k}{\tau} = \frac{\varepsilon}{16\log(k/\varepsilon)} \log \left( \frac{16k}{\varepsilon/\log(k/\varepsilon)} \right) \leqslant \frac{1}{8}\varepsilon,$$

when $\tau \leqslant 1/e$. Similarly, if $p(\bar{\mathcal{A}}) \leqslant 3\tau$, we have that $H(p_{\bar{\mathcal{A}}}) \leqslant \frac{3}{8}\varepsilon$. Therefore,

$$
\begin{aligned}
\varepsilon \leqslant |H(p) - H(q)| &\leqslant |H(p_{\mathcal{A}}) - H(q_{\mathcal{A}})| + |H(p_{\bar{\mathcal{A}}}) - H(q_{\bar{\mathcal{A}}})| \\
&\leqslant |H(p_{\mathcal{A}}) - H(q_{\mathcal{A}})| + |H(p_{\bar{\mathcal{A}}})| + |H(q_{\bar{\mathcal{A}}})| \\
&\leqslant |H(p_{\mathcal{A}}) - H(q_{\mathcal{A}})| + \frac{1}{2}\varepsilon.
\end{aligned}
$$

For Line 4, we prove the following: with probability at least $99/100$, if $Z_1 \geqslant 2\tau$, then $p(\bar{\mathcal{A}}) \geqslant \tau$; and if $Z_1 < 2\tau$, then $p(\bar{\mathcal{A}}) < 3\tau$ (this is a standard technique; see e.g., [Can22, Fact 2.2].) For the sake of completeness we include the full derivation in the Appendix A.

**After Line 4 of Algorithm 1.** We conclude from the above that

    i. $\mathcal{A}$ still has sufficient entropy gap to test on: $|H(p_{\mathcal{A}}) - H(q_{\mathcal{A}})| \geqslant \frac{1}{2}\varepsilon$.

    ii. With probability at least $99/100$, when $p = q$, it will not be rejected in Algorithm 4 of Line 4; and once it is pass through this stage, we have $p(\bar{\mathcal{A}}) \leqslant 3\tau$.

**Completeness: when $p = q$.**

- We have that $d_{\chi^2}(p_{\mathcal{A}}, q_{\mathcal{A}}) = 0$, and via Lemma 2.4, we know that $\mathbb{E}[Z_2] = 0$ and $\mathrm{Var}[Z_2] \leqslant \frac{1}{32^2} m_2^2 \varepsilon^2$. By Chebyshev's inequality,

$$\Pr\left[ |Z_2 - \mathbb{E}[Z_2]| \geqslant 2\sqrt{\mathrm{Var}[Z_2]} \right] \leqslant \frac{1}{4}, \quad \text{and so} \quad \Pr[Z_2 \geqslant 2 \cdot \frac{1}{32} m_2 \varepsilon + \mathbb{E}[Z_2]] \leqslant \frac{1}{4};$$

and we have $\Pr[Z_2 \geqslant \frac{1}{16} m_2 \varepsilon] \leqslant \frac{1}{4}$.

- On the other hand, by Claim 2.3, setting $m_3 = \frac{140800 \log^2(k)}{\varepsilon^2}$, we have that with probability at least $99/100$,

$$Z_3 = \left| \sum_{i \in \mathcal{A}} (\hat{p}_i - q_i) \log \frac{1}{q_i} \right| = \left| \sum_{i \in \mathcal{A}} (\hat{p}_i - p_i) \log \frac{1}{q_i} \right| \leqslant \frac{1}{8} \varepsilon.$$

Therefore, with probability at least $1 - \frac{1}{4} - \frac{2}{100} = \frac{73}{100} > \frac{2}{3}$, the tester will accept.

**Soundness: when** $|H(p) - H(q)| \geqslant \varepsilon$. If $p(\bar{\mathcal{A}}) \geqslant 3\tau$ then $\hat{p}(\bar{\mathcal{A}}) \geqslant 2\tau$ with probability $99/100$, and the algorithm will output Reject. We proceed assuming $p(\bar{\mathcal{A}}) \leqslant 3\tau$ and recall Item ii. from before, we have $|H(p_{\mathcal{A}}) - H(q_{\mathcal{A}})| \geqslant \frac{1}{2}\varepsilon$. By Claim 2.2, we have that either $d_{\mathrm{KL}}(p_{\mathcal{A}}, q_{\mathcal{A}}) \geqslant \frac{1}{4}\varepsilon$ or $\left| \sum_{i \in \mathcal{A}} (p_i - q_i) \log(1/q_i) \right| \geqslant \frac{1}{4}\varepsilon$. We apply Lemma 2.4, setting $\alpha = \tau/k$ and $m_2 \geqslant 65536 \sqrt{k \log(k/\varepsilon)}/\varepsilon$.

- If $d_{\mathrm{KL}}(p_{\mathcal{A}} \| q_{\mathcal{A}}) \geqslant \frac{1}{4}\varepsilon$, with (4) and $\exp(3/2) \leqslant k/\varepsilon$, we have

$$\frac{1}{8}\varepsilon \leq -3\tau + d_{\mathrm{KL}}(p_{\mathcal{A}}, q_{\mathcal{A}}) \leqslant \sum_{i \in \mathcal{A}} (q_i - p_i) + d_{\mathrm{KL}}(p_{\mathcal{A}}, q_{\mathcal{A}}) \leqslant d_{\chi^2}(p_{\mathcal{A}}, q_{\mathcal{A}}),$$

which by Lemma 2.4, and our setting of $m_2$ and $\alpha$, implies $\mathrm{Var}[Z_2] \leqslant (\frac{1}{4}\mathbb{E}[Z_2])^2$ and $\mathbb{E}[Z_2] = m_2 \cdot d_{\chi^2}(p_{\mathcal{A}}, q_{\mathcal{A}}) \geqslant \frac{1}{8}m_2\varepsilon$. By Chebyshev,

$$\Pr\left[ |Z_2 - \mathbb{E}[Z_2]| \geqslant 2\sqrt{\mathrm{Var}[Z_2]} \right] \leqslant \frac{1}{4} \text{ and so } \Pr[Z_2 \leqslant \frac{1}{16}m_2\varepsilon] \leqslant \frac{1}{4}.$$

- On the other hand, if it is the case that $\left| \sum_{i \in \mathcal{A}} (p_i - q_i) \log(1/q_i) \right| \geqslant \frac{1}{4}\varepsilon$, by Claim 2.3, setting $m_3 = 140800 \log^2(k)/\varepsilon^2$, with probability at least $99/100$,

$$
\begin{aligned}
\frac{1}{4}\varepsilon &\leqslant \left| \sum_i p_i \log \frac{1}{q_i} - q_i \log \frac{1}{q_i} \right| \\
&\leqslant \left| \sum_i (p_i - \hat{p}_i) \log \frac{1}{q_i} \right| + \left| \sum_i (\hat{p}_i - q_i) \log \frac{1}{q_i} \right| \\
&\leqslant \frac{1}{8}\varepsilon + \left| \sum_i (\hat{p}_i - q_i) \log \frac{1}{q_i} \right|.
\end{aligned}
$$

We have that $Z_3 = \left| \sum_i (\hat{p}_i - q_i) \log \frac{1}{q_i} \right| \geqslant \frac{1}{8}\varepsilon$ and thus with probability at least $1 - \frac{1}{4} - \frac{2}{100} = \frac{73}{100}$, the following will happen, the tester will reject: either $p(\bar{\mathcal{A}}) \geqslant 3\tau$, and it is rejected at Line 4 of Algorithm 4, or it passes and $p(\bar{\mathcal{A}}) \leqslant 3\tau$ and

$$Z_2 \geqslant \frac{1}{8}m_2\varepsilon \text{ or } Z_3 \geqslant \frac{1}{8}\varepsilon,$$

and will be rejected. This concludes the proof. □

**Remark 2.5.** *We note that we can slightly improve the sample complexity of Theorem 1.1 (specifically, improving on the $\sqrt{k \log(k/\varepsilon)}$ term), at the price of a more complicated algorithm, by adding thresholds $\tau' = \frac{\varepsilon}{\log \log(k/\varepsilon)}$, $\tau'' = \frac{\varepsilon}{\log \log \log(k/\varepsilon)}$, and considering separately the elements in $\mathcal{A}' = \{i : q_i \in (\tau/k, \tau'/k]\}$, $\mathcal{A}'' = \{i : q_i \in (\tau'/k, \tau''/k]\}$; specifically, by grouping them in groups, and "merging" each group to get a "new" element with larger probability. For the sake of clarity, we defer this improvement to Appendix D.*

## 2.2 An $\Omega(\sqrt{k}/\varepsilon + \log^2 k/\varepsilon^2)$ lower bound

The $\Omega(\sqrt{k}/\varepsilon + \log^2 k/\varepsilon^2)$ lower bound comes from the combination of Lemma 2.6 and Lemma 2.7. We obtain Lemma 2.6 through the classical hard instance used for uniformity testing [Pan08] and a simple conversion between TV distance and entropy difference gives the result. We note that distributions close to uniform distribution actually have smaller entropy difference (uniform distribution is quite special: having the highest entropy of $\log k$). Indeed, replacing the uniform distribution with a slightly biased distribution, we obtain another hard instance for Lemma 2.7, using the classical Le Cam's two-point method.

**Algorithm 2** Identity testing for bounded degree Bayes nets

---

**Require:** Sample access to Bayes net $p$, full description of Bayes net $q$, accuracy parameter $\varepsilon$, in-degree $d$ and dimension $n$.

1: $\mathcal{S}_1 \leftarrow O\left(\left(\frac{2^{d/2}n\sqrt{d\log(n/\varepsilon)}}{\varepsilon^2} + \frac{d^2 n^2}{\varepsilon^4}\right) d\log n\right)$ samples from $p$;

2: $\mathcal{S}_2 \leftarrow O\left(\frac{2^{d/2}n}{\varepsilon^2}\sqrt{\log(1/\varepsilon)} \cdot \log n\right)$ samples from $p$;

3: **for all** $L \in \mathcal{N}_{d+1} \cup \mathcal{N}_d$ **do** $\qquad\qquad\qquad \triangleright \mathcal{N}_\ell$ is all subsets of $\{0,1\}^n$ with size $\ell$

4: $\quad$ Call Algorithm 1 with $p_L, q_L$ and $\mathcal{S}_1$; $\quad \triangleright$ Entropy test on $p_L$ and $q_L$ with accuracy $\varepsilon^2/n$.

5: $\quad$ **if** Entropy test rejects **then return** reject

6: $S_3 \leftarrow O\left(\frac{n^2 \cdot \log(n/\varepsilon) \cdot d\log(n)}{\varepsilon^2}\right)$ samples from $p$ and compute its empirical distribution $\hat{p}$;

7: **for all** $i \in [n]$ **do**

8: $\quad$ **if** $\hat{p}_{X_i,\Pi_i^G}(\bar{\mathcal{A}}_i') \geqslant \Omega(\varepsilon^2/(n^2 \cdot \log(n/\varepsilon)))$ **then return** reject

9: $\quad$ Check $d_{\mathrm{KL}}(p_{X_i,\Pi_i^G;\mathcal{A}_i'}\|q_{X_i,\Pi_i^G;\mathcal{A}_i'}) \geqslant \frac{\varepsilon^2}{n}$ or $d_{\mathrm{KL}}(p_{X_i,\Pi_i^G;\mathcal{A}_i'}\|q_{X_i,\Pi_i^G;\mathcal{A}_i'}) = 0$.

10: $\quad$ **if** $i$-th KL test says far **then return** reject

11: **return** accept $\qquad\qquad\qquad\qquad\qquad\qquad\qquad\qquad\qquad \triangleright$ Accept if all tests pass.

---

**Lemma 2.6.** *With fewer than $c_3 \cdot \sqrt{k}/\varepsilon$ samples from $p$, no tester can distinguish between $p = q$ and $|H(p) - H(q)| \geqslant \varepsilon$ with probability higher than $2/3$, where $c_3 > 0$ is an absolute constant.*

**Lemma 2.7.** *With fewer than $c_4 \cdot \log^2 k/\varepsilon^2$ samples from $p$, no tester can distinguish between $p = q$ and $|H(p) - H(q)| \geqslant \varepsilon$ with probability higher than $2/3$, where $c_4 > 0$ is an absolute constant.*

## 3 Application to identity testing for Bayes nets

We now provide an application of our main entropy identity testing theorem, to obtain an improved "standard" identity testing algorithm for Bayesian networks:

**Theorem 3.1.** *Given sample access to an in-degree $d$ Bayes net $p$ and full description of in-degree $d$ Bayes net $q$, Algorithm 2 takes*

$$C \cdot \left(\frac{2^{d/2}nd^{3/2}\log n \cdot \sqrt{\log(n/\varepsilon)}}{\varepsilon^2} + \frac{d^3 n^2 \cdot \log n}{\varepsilon^4} + \frac{2^{d/2}n^{3/2} \cdot \log n}{\varepsilon^2}\right)$$

*samples to test between $p = q$ or $\mathrm{d_H}(p, q) \geqslant \Omega(\varepsilon)$, where $C > 0$ is an absolute constant. Moreover, the algorithm runs in time polynomial in $n^d$ and $1/\varepsilon$.*

Before proceeding to the analysis of our algorithm, we require the following definitions.

**Definition 3.2.** *A projection of a Bayes net $p$ on $\{0,1\}^n$ unto a DAG $G$ is denoted $p_G$, and is defined by its probability mass function (PMF) as follows:*

$$p_G(X_1, \ldots, X_n) = \prod_{i=1}^n p(X_i \mid \Pi_i^G),$$

*where $\Pi_i^G$ is the set of parents of $X_i$ in $G$. Abusing the notation in the context of Bayesian networks, we refer to $p_{X_i,\Pi_i}$ or $p_{X_i,\Pi_i}(x_i, \pi_i)$ as the marginal distribution of $p$ on the subset $\{X_i, \Pi_i\}$.*

Denote $\mathcal{U} := \bigcup_{i=1}^n \mathcal{A}_i$, where $\mathcal{A}_i := \left\{x \in \{0,1\}^n : q_{X_i,\Pi_i^G}(x_i(x), \pi_i^G(x)) \geqslant \Omega\left(\frac{\varepsilon^2}{2^{d+1}n^2\log(n/\varepsilon)}\right)\right\}$. This gives us the property that marginalization over $X_i = x_i, \Pi_i^G = \pi_i^G$ works nicely as we include elements only based on its local property (as long as $q_{X_i,\Pi_i}$ is large enough). And $q$ is Markov w.r.t. $G$. We use $(x_i, \pi_i) \in \mathcal{A}_i'$, where $\mathcal{A}_i' = \left\{x' \in \{0,1\}^{|\Pi_i^G|+1} : q_{X_i,\Pi_i^G}(x') \geqslant \Omega\left(\frac{\varepsilon^2}{2^{d+1}n^2\log(n/\varepsilon)}\right)\right\}$. Let $(a, B) \in \mathcal{A}_i'$, we have that as long as $(x_i(x), \pi_i(x)) = (a, B)$, then $x \in \mathcal{A}_i$ and vice versa, which means that

$$\mathcal{U} = \bigcup_{i=1}^n \mathcal{A}_i = \bigcup_{i=1}^n \{x \in \{0,1\}^n : (x_i(x), \pi_i^G(x)) \in \mathcal{A}_i'\}.$$

We will check if $p_{X_i,\Pi_i^G}(\bar{\mathcal{A}}_i') \geqslant \Omega(\varepsilon^2/(n^2 \cdot \log(n/\varepsilon)))$ and reject early if true; this takes $O\left(\frac{n^2 \cdot \log(n/\varepsilon) \cdot d \log(n)}{\varepsilon^2}\right)$ samples for all tests to be correct via a union bound. After passing this test, we can conclude that

$$p(\bar{\mathcal{U}}) = \sum_{x \in \bigcap_{i=1}^n \bar{\mathcal{A}}_i} p(x) \leqslant \sum_{x \in \bar{\mathcal{A}}_1} p(x) = \sum_{x' \in \bar{\mathcal{A}}_1'} p_{X_1,\Pi_1^G}(x') = p_{X_1,\Pi_1^G}(\bar{\mathcal{A}}_1') \leqslant O(\frac{\varepsilon^2}{n^2 \cdot \log(n/\varepsilon)}),$$

where we marginalize over everything other than $(X_1, \Pi_1^G)$ in the third step.

Similarly, we can upper bound $q(\bar{\mathcal{U}}) \leqslant q_{X_i,\Pi_i^G}(\bar{\mathcal{A}}_i') \leqslant O(\varepsilon^2/(n^2 \cdot \log(n/\varepsilon)))$. Abusing the notation slightly, we denote $p_{G;\mathcal{U}}$ as the distribution obtained by projecting $p$ onto $G$ (which gives $p_G$) and then restricting the distribution $p_G$ to take elements in $\mathcal{U}$.

We will need the following Lemma 3.3 whose proof is deferred to Appendix E.

**Lemma 3.3.** *Suppose* $d_H^2(p,q) \geqslant \Omega(\varepsilon^2)$; $d_{\mathrm{KL}}(p\|p_G) \leqslant O(\varepsilon^2)$; $p(\bar{\mathcal{U}}) \leqslant \frac{\varepsilon^2}{n \log(n/\varepsilon)}$ ; $\forall i \in [n], p(\bar{\mathcal{A}}_i') \leqslant \frac{\varepsilon^2}{n^2 \log(n/\varepsilon)}$, *where* $\mathcal{A}_i'$ *is defined above, and* $q$ *is Markov with respect to* $G$, *then we have*

$$\sum_{i=1}^n d_{\mathrm{KL}}(p_{X_i,\Pi_i^G;\mathcal{A}_i'}\|q_{X_i,\Pi_i^G;\mathcal{A}_i'}) \geqslant \Omega(\varepsilon^2).$$

*Therefore testing* $d_{\mathrm{KL}}(p_{X_i,\Pi_i^G;\mathcal{A}_i'}\|q_{X_i,\Pi_i^G;\mathcal{A}_i'}) \geqslant \frac{\varepsilon^2}{n}$ *over all* $i$ *suffices to detect this case.*

In addition, to connect entropy testing to Bayes net testing, we will need the following Lemma 3.4, whose proof will be deferred to Appendix E. And so if we can show that all local entropies between Bayes nets $p$ and $q$ are sufficiently close, then this implies that $p$ must be close to $q$'s DAG $G$ and $q$ must also be close to $p$'s DAG $G'$.

**Lemma 3.4.** *Let* $p$ *and* $q$ *be two max in-degree-*$d$ *Bayes nets supported on* $\{0,1\}^n$ *such that for every subset* $L \subseteq \{X_1, \cdots, X_n\}$ *of size* $d+1$, *the following holds:*

$$|H(p_L) - H(q_L)| \leqslant O\left(\frac{\varepsilon^2}{n}\right).$$

*Suppose* $p$ *is Markov w.r.t.* $G'$ *and* $q$ *Markov w.r.t.* $G$. *Then we have that*

$$d_{\mathrm{KL}}(p\|p_G) \leqslant O(\varepsilon^2) \text{ and } d_{\mathrm{KL}}(q\|q_{G'}) \leqslant O(\varepsilon^2).$$

*Proof of Theorem 3.1.* We show the result by analyzing Algorithm 2. By Theorem 1.1, the sample complexity for entropy testing on any subset $L$ of size (dimension) $d$ or $d+1$, is

$$O\left(2^{d/2}n\sqrt{d\log(n/\varepsilon)}/\varepsilon^2 + d^2n^2/\varepsilon^4\right).$$

To guarantee the success of every tests employed in the algorithm, we increase the sample complexity of each test by an extra $O(\log(n^{d+1})) = O(d \log n)$ factor to boost their success probability to $1 - \frac{1}{100n^{d+1}}$ (via a standard majority vote technique), which will allow us to use a union bound over all tests as there are at most $n^{d+1}$ subsets with size $d+1$. For this step, the sample complexity will be

$$O\left(\left(\frac{2^{d/2}n\sqrt{d\log(n/\varepsilon)}}{\varepsilon^2} + \frac{d^2n^2}{\varepsilon^4}\right)d\log n\right).$$

With this in hand, we will proceed with the analysis under the event that every entropy test performed is correct (which by the above argument happens with high probability). If distribution $p$ manages to pass all the entropy tests, it must satisfy the following:

$$|H(p_L) - H(q_L)| \leqslant \frac{\varepsilon^2}{n}, \tag{6}$$

for every subset $L$ of size $d+1$ for the latter, and every subset $L$ of size $d$ or $d+1$ for the former. From here, in principle, we can perform structural learning of $p$ through $H(q_L)$, which then gives us an approximated DAG $\hat{G}$ of $p$ and we can check $d_H(p_{\hat{G}}, q_{\hat{G}})$. Unfortunately, structural learning

of Bayes nets is known to be computationally hard in many settings [Höf93, CHM04], and so this would lead to a computationally inefficient algorithm.

Instead, we argue that this (learning) step can be bypassed entirely: the intuition of the argument is to view structure learning for Bayes net as an optimization problem; and any assignment $x$ to the two optimization problems (structure learning of $p$ and $q$) satisfy $f_1(x) = f_2(x) \pm O(\varepsilon^2)$ due to their local entropy being close[7] – this means that an optima $x_1$ for $f_1$ satisfies $\min_x f_1(x) = f_1(x_1) \geqslant f_2(x_1) - \varepsilon^2 \geqslant \min_x f_2(x) - \varepsilon^2$ and vice versa (optima $x_2$ for $f_2$).

Applying Lemma 3.4, we have that $d_{\mathrm{KL}}(p\|p_G) \leqslant O(\varepsilon^2)$ where $G$ is the DAG $q$ is Markov with respect to. With this at hand, we continue onto the KL testing part. The algorithm will check if $p_{X_i,\Pi_i^G}(\bar{\mathcal{A}}_i') \geqslant \Omega(\varepsilon^2/\max(n, \log(1/\varepsilon)))$ and reject early if it is true (this costs $O\left(\frac{d\log(n)\cdot\max(n,\log(1/\varepsilon))}{\varepsilon^2}\right)$ samples) and then check for every $i \in [n]$,

$$d_{\mathrm{KL}}(p_{X_i,\Pi_i^G;\mathcal{A}_i'}\|q_{X_i,\Pi_i^G;\mathcal{A}_i'}) \geqslant \frac{\varepsilon^2}{n} \ \text{ or } \ d_{\mathrm{KL}}(p_{X_i,\Pi_i^G;\mathcal{A}_i'}\|q_{X_i,\Pi_i^G;\mathcal{A}_i'}) = 0.$$

Recalling (4), if the former case holds, we have

$$d_{\chi^2}(p_{X_i,\Pi_i^G;\mathcal{A}_i'}\|q_{X_i,\Pi_i^G;\mathcal{A}_i'}) \geqslant \underbrace{p_{X_i,\Pi_i^G}(\bar{\mathcal{A}}_i') - q_{X_i,\Pi_i^G}(\bar{\mathcal{A}}_i')}_{\leqslant O(\varepsilon^2/n)} + d_{\mathrm{KL}}(p_{X_i,\Pi_i^G;\mathcal{A}_i'}\|q_{X_i,\Pi_i^G;\mathcal{A}_i'}) = \Omega\left(\frac{\varepsilon^2}{n}\right)$$

the bound on the first term following from the algorithm's check on Line 10. Using Lemma 2.4, we can perform the corresponding check for $i \in [n]$, and after a union bound over $n$ tests, noting that $q(x_i, \pi_i^G) \geqslant \frac{\varepsilon^2}{2^{d+1}n^2\log(n/\varepsilon)}$ when restricted on $\mathcal{A}_i'$, the sample complexity is

$$O\left(\left(\sqrt{1/\left(\frac{\varepsilon^2}{2^{d+1}n^2\log(n/\varepsilon)}\cdot\frac{\varepsilon^2}{n}\right)} + \frac{\sqrt{2^{d+1}}\cdot n}{\varepsilon^2}\right)\cdot\log n\right) = O\left(\frac{2^{d/2}n^{3/2}}{\varepsilon^2}\cdot\log n\right).$$

Following this, we look at the two cases:

- If $p = q$, then with high probability, $p$ will pass all entropy tests, all KL local tests and the tester will accept.
- If $d_{\mathrm{H}}(p, q) \geqslant \varepsilon$, either it fails one of the entropy tests. If it does pass the entropy test, then we must have that $d_{\mathrm{KL}}(p\|p_G) \leqslant O(\varepsilon^2)$ by (26). Then following Lemma 3.3 and Lemma 2.4, the tester will reject.

In total, the sample complexity is:

$$O\left(\left(\frac{2^{d/2}n\sqrt{d\log(n/\varepsilon)}}{\varepsilon^2} + \frac{d^2n^2}{\varepsilon^4}\right)d\log n + \frac{2^{d/2}n^{3/2}}{\varepsilon^2}\cdot\log n\right).$$

This concludes the proof of the theorem. □

## 4 Conclusion and open problems

In this paper, we study a variant of distribution testing problem in terms of entropy difference; we give nearly tight upper and lower sample complexity bounds for the problem. We subsequently apply our entropy testing algorithm to identity testing of Bayes nets, which unlike prior works, makes merely the necessary assumptions (the bound on the in-degree of the Bayes nets).

**Future directions.** We believe the *closeness* (two-sample) testing variant of the problem (testing if two unknown distribution $p$ and $q$ are the same or far in terms of entropy difference) could also be interesting; and, notably, has connections to other distribution testing problems: first, it should lead to a natural solution to closeness testing of Bayes nets via ideas in this paper. Second, solving the closeness entropy testing problem give another path to testing independence in terms of mutual information (studied in [BGP+23] and also covered in [CDKS18]), a notion closely related to entropy.

---

[7] Here, $x$ is the DAG's assignment of parents and child; and $f_1(x)$ (resp. $f_2(x)$) is the associated KL-divergence (also called *score* of the DAG in the literature, which measures the how well DAG models the true Bayes net) between $p$ (resp. $q$) and $x$. Since we are in the realizable setting, the optimal is in fact 0.

## Acknowledgments and Disclosure of Funding

We would like to thank the reviewers for their suggestions and efforts which help improve this paper.

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

## A    Derivation for Line 4 in Algorithm 1

We need to show the following: with probability at least $99/100$, if $Z_1 \geqslant 2\tau$, then $p(\bar{A}) \geqslant \tau$; and if $Z_1 < 2\tau$, then $p(\bar{A}) < 3\tau$. For the first one, we prove by contrapositive: with high probability $1 - \frac{1}{200}, p(\bar{A}) < \tau \Rightarrow Z_1 < 2\tau$. Suppose $T = \text{Binomial}(m_1, \tau)$ and setting $m_1 = \frac{48}{\tau} \geqslant \frac{3\log 200}{\tau}$, and using a Chernoff bound, we have the following

$$\Pr[T \geqslant 2\tau] \leqslant \exp(-\tau \cdot m_1/3) \leqslant \frac{1}{200}.$$

Since any $\text{Binomial}(m_1, p(\bar{A}))$ will be first-order stochastic dominated by $\text{Binomial}(m_1, \tau)$ if $p(\bar{A}) < \tau$, we can conclude the following: if $p(\bar{A}) < \tau$, then $\Pr[Z_1 \geqslant 2\tau] \leqslant \Pr[T \geqslant 2\tau] \leqslant \frac{1}{200}$.

For the latter, we prove via its contrapositive: with probability $1 - \frac{1}{200}, p(\bar{A}) \geqslant 3\tau \Rightarrow Z_1 \geqslant 2\tau$. As $p(\bar{A}) \geqslant 3\tau$, take $m_1 = \frac{48}{\tau} \geqslant \frac{9\log 200}{\tau}$, by a Chernoff bound, we have

$$\Pr[\hat{p}'(\bar{A}) \leqslant 2\tau] \leqslant \Pr[\hat{p}'(\bar{A}) \leqslant (1-1/3) \cdot p(\bar{A})] \leqslant \exp\left(-\frac{m_1 \cdot p(\bar{A})}{18}\right) \leqslant \exp\left(-\frac{m_1 \cdot \tau}{9}\right) \leqslant \frac{1}{200}.$$

Combining the two with a union bound concludes the proof.

## B    Deferred proofs from Section 2.1

**Claim 2.2.** *Let $A$ be any set such that $p(\bar{A}) < \varepsilon/2$. Then, if $|H(p_A) - H(q_A)| \geqslant \varepsilon$, we must have (i) $d_{\text{KL}}(p_A \| q_A) \geqslant \frac{\varepsilon}{2}$ or (ii) $|\sum_{i \in A}(p_i - q_i)\log(\frac{1}{q_i})| \geqslant \frac{\varepsilon}{2}$.*

*Proof of Claim 2.2.* We can bound $|H(p_A) - H(q_A)|$ as

$$
\begin{aligned}
\varepsilon \leq |H(p_A) - H(q_A)| \quad &= \quad \left|\sum_{i \in A}\left(p_i \log \frac{1}{p_i} - q_i \log \frac{1}{q_i}\right)\right| \\
&= \quad \left|\sum_{i \in A}\left(p_i \log \frac{q_i}{p_i} + p_i \log \frac{1}{q_i} - q_i \log \frac{1}{q_i}\right)\right| \\
&\leqslant \quad \left|\sum_{i \in A} p_i \log \frac{q_i}{p_i}\right| + \left|\sum_{i \in A}\left(p_i \log \frac{1}{q_i} - q_i \log \frac{1}{q_i}\right)\right| \\
&= \quad |d_{\text{KL}}(p_A \| q_A)| + \left|\sum_{i \in A}(p_i - q_i)\log \frac{1}{q_i}\right|, \quad\quad\quad (7)
\end{aligned}
$$

which implies that at least one of the two terms is at least $\varepsilon/2$. If it is the second, we are done; otherwise, we know that either

$$d_{\text{KL}}(p_A \| q_A) \geqslant \frac{1}{2}\varepsilon \text{ or } d_{\text{KL}}(p_A \| q_A) \leqslant -\frac{1}{2}\varepsilon.$$

We will rule out the second case, using that $\log \frac{1}{x} \geqslant 1 - x$ for $x > 0$,[8]

$$d_{\text{KL}}(p_A \| q_A) = \sum_{i \in A} p_i \log \frac{p_i}{q_i} \geqslant \sum_{i \in A} p_i \left(1 - \frac{q_i}{p_i}\right) = q(\bar{A}) - p(\bar{A}) > -\frac{1}{2}\varepsilon.$$

Thus, we cannot have $d_{\text{KL}}(p_A \| q_A) \leqslant -\frac{1}{2}\varepsilon$ and so $d_{\text{KL}}(p_A \| q_A) \geqslant \frac{1}{2}\varepsilon$.    $\square$

**Claim 2.3.** *Let $\hat{p}$ be the empirical estimator for an unknown discrete distribution $p$ supported on $[k]$, based on $\text{Poi}(m)$ samples, where $m = \Theta\left(\frac{\log^2(k)}{\varepsilon^2}\right)$; assume that $d_{\chi^2}(p_A, q_A) \leqslant \varepsilon/8$ and $p(\bar{A}) + q(\bar{A}) \leqslant 4\tau = \frac{1}{4}\frac{\varepsilon}{\log(k/\varepsilon)},$[9] then*

$$\Pr\left[\left|\sum_{i \in A}(p_i - \hat{p}_i)\log \frac{1}{q_i}\right| \geqslant \frac{1}{8}\varepsilon\right] \leqslant \frac{1}{100}.$$

---

[8] In the case of $p_i = 0$, we still have $p_i \log\left(\frac{p_i}{q_i}\right) \geqslant p_i - q_i$.

[9] One can remove the assumption that $p(\bar{A}) + q(\bar{A}) \leqslant 4\tau$, at the cost of a slightly worse constant.

*Proof of Claim 2.3.* We follow the same analysis as in [WY16]. Letting $Y_i := (p_i - \hat{p}_i) \log \frac{1}{q_i}$ for $i \in \mathcal{A}$, we have $\mathbb{E}[Y_i] = 0$ and

$$\text{Var}[Y_i] = \mathbb{E}[(Y_i - \mathbb{E}[Y_i])^2] = \mathbb{E}[Y_i^2] = \mathbb{E}[(p_i - \hat{p}_i)^2] \log^2 \frac{1}{q_i} = \frac{1}{m^2}(mp_i) \log^2 \frac{1}{q_i} = \frac{p_i}{m} \log^2 \frac{1}{q_i}.$$

Let $Y := \sum_{i \in \mathcal{A}}(p_i - \hat{p}_i) \log \frac{1}{q_i}$. We will use our assumption that $d_{\chi^2}(p_{\mathcal{A}}, q_{\mathcal{A}}) \leqslant \varepsilon/8$ and $p(\bar{\mathcal{A}}) + q(\bar{\mathcal{A}}) \leqslant \frac{1}{4} \frac{\varepsilon}{\log(k/\varepsilon)}$ below. Note that, our analysis is in the Poissonized setting:

$$
\begin{aligned}
\text{Var}[Y] \;=\;& \text{Var}\left[\sum_{\mathcal{A}}(p_i - \hat{p}_i) \log \frac{1}{q_i}\right] \\
=\;& \sum_{i \in \mathcal{A}} \frac{p_i}{m} \log^2\left(\frac{1}{q_i}\right) \\
=\;& \sum_{i \in \mathcal{A}} \frac{p_i}{m}\left(\log\left(\frac{1}{p_i}\right) + \log\left(\frac{p_i}{q_i}\right)\right)^2 \\
\leqslant\;& \sum_{i \in \mathcal{A}} \frac{p_i}{m}\left(2\left(\log\left(\frac{1}{p_i}\right)\right)^2 + 2\left(\log\left(\frac{p_i}{q_i}\right)\right)^2\right) \\
=\;& \sum_{i \in \mathcal{A}} \frac{2}{m} p_i \log^2\left(\frac{1}{p_i}\right) + \sum_{i \in \mathcal{A}} \frac{2}{m} p_i \log^2\left(\frac{p_i}{q_i}\right) \\
\leqslant\;& \sum_{i \in \mathcal{A}} \frac{2}{m} p_i \log^2\left(\frac{1}{p_i}\right) + \sum_{\frac{p_i}{q_i} \geqslant 1, i \in \mathcal{A}} \frac{2}{m} p_i\left(\frac{p_i}{q_i} - 1\right) + \sum_{\frac{p_i}{q_i} < 1, i \in \mathcal{A}} \frac{2}{m} p_i\left(\frac{q_i}{p_i} - 1\right) \quad (8) \\
=\;& \sum_{i \in \mathcal{A}} \frac{2}{m} p_i \log^2\left(\frac{1}{p_i}\right) + \sum_{\frac{p_i}{q_i} \geqslant 1, i \in \mathcal{A}} \frac{2}{m} \frac{(p_i - q_i)^2}{q_i} + \sum_{\frac{p_i}{q_i} \geqslant 1, i \in \mathcal{A}} \frac{2}{m}(p_i - q_i) + \sum_{\frac{p_i}{q_i} < 1, i \in \mathcal{A}} \frac{2}{m}(q_i - p_i) \\
\leqslant\;& \frac{4 \log^2 k}{m} + \frac{6}{m} + \frac{2}{m}(d_{\chi^2}(p_{\mathcal{A}}, q_{\mathcal{A}}) + d_{\text{TV}}(p, q)) \quad (9) \\
\leqslant\;& \frac{4 \log^2 k}{m} + \frac{6}{m} + \frac{2}{m}\left(\frac{\varepsilon}{8} + \sqrt{\frac{\varepsilon}{8}} + 4\tau\right) \leqslant \frac{4 \log^2 k}{m} + \frac{8}{m} \leqslant \frac{22 \log^2 k}{m} \quad (10)
\end{aligned}
$$

For (8), we analyze by two cases: if $\frac{p_i}{q_i} \geqslant 1$, we have that $p_i \log^2\left(\frac{p_i}{q_i}\right) \leqslant p_i\left(\frac{p_i}{q_i} - 1\right)$; otherwise, $p_i \log^2\left(\frac{p_i}{q_i}\right) = p_i \log^2\left(\frac{q_i}{p_i}\right) < p_i\left(\frac{q_i}{p_i} - 1\right)$. And we use [HJW15b, Lemma3], $\sum_{i \in \mathcal{A}} p_i \log^2\left(\frac{1}{p_i}\right) \leqslant 2 \log^2 k + 3$ in (9). We use the premise and (4) in (10) and we have that

$$d_{\text{TV}}(p, q) = d_{\text{TV}}(p_{\mathcal{A}}, q_{\mathcal{A}}) + d_{\text{TV}}(p_{\bar{\mathcal{A}}}, q_{\bar{\mathcal{A}}}) \leqslant \sqrt{d_{\chi^2}(p_{\mathcal{A}}, q_{\mathcal{A}})} + p(\bar{\mathcal{A}}) + q(\bar{\mathcal{A}}) \leqslant \sqrt{\frac{\varepsilon}{8}} + 4\tau;$$

and the last step is obtained by noticing that $\log(k) \geqslant \frac{2}{3}$ for $k \geqslant 2$. By Chebyshev's inequality, we then have that

$$\Pr\left[|Y| \geqslant 10\sqrt{\frac{38 \log^2 k}{m}}\right] \leqslant \Pr\left[|Y - \mathbb{E}[Y]| \geqslant 10\sqrt{\text{Var}[Y]}\right] \leqslant \frac{1}{100},$$

and this last inequality yields the statement as long as $m \geqslant \frac{22 \times 100 \times 8^2 \log^2(k)}{\varepsilon^2} = \frac{140800 \log^2(k)}{\varepsilon^2}$. $\quad \square$

**Lemma 2.4.** *Let $\mathcal{A} := \{i \in [k] \mid q_i \geqslant \alpha\}$. Let $m_2 \geqslant 16384 \max\left\{\sqrt{\frac{1}{\alpha\varepsilon}}, \frac{\sqrt{k}}{\varepsilon}\right\}$ be the number of samples used to compute $Z_2$. Then $\mathbb{E}[Z_2] = m_2 d_{\chi^2}(p_{\mathcal{A}}, q_{\mathcal{A}})$. Moreover, if $d_{\chi^2}(p_{\mathcal{A}}, q_{\mathcal{A}}) \leqslant \frac{\varepsilon}{2}$, then $\text{Var}[Z_2] \leqslant \left(\frac{1}{32} m_2 \varepsilon\right)^2$. If $d_{\chi^2}(p_{\mathcal{A}}, q_{\mathcal{A}}) \geqslant \varepsilon$, then $\text{Var}[Z_2] \leqslant O(\mathbb{E}[Z_2]^2)$.*

*Proof of Lemma 2.4.* The proof is a relatively straightforward modification of the argument of [DKW18, Lemma 2]. We have the expectation and variance of $Z_2$,

$$\mathbb{E}[Z_2] = m_2 d_{\chi^2}(p_{\mathcal{A}}, q_{\mathcal{A}}) \text{ and } \operatorname{Var}[Z_2] = \sum_{i \in \mathcal{A}} \left[ 2\frac{p_i^2}{q_i^2} + 4m_2 \frac{p_i(p_i - q_i)^2}{q_i^2} \right].$$

It boils down to bounding the following,

$$
\begin{aligned}
2\sum_{i \in \mathcal{A}} \frac{p_i^2}{q_i^2} &\leqslant 4k + 4\sum_{i \in \mathcal{A}} \frac{(p_i - q_i)^2}{q_i^2} \\
&\leqslant 4k + \frac{4}{\alpha} \sum_{i \in \mathcal{A}} \frac{(p_i - q_i)^2}{q_i} \\
&\leqslant 4k + \frac{4}{\alpha m_2} \mathbb{E}[Z_2].
\end{aligned}
$$

Derivation of the inequalities follow from [DKW18, proof of Lemma 2].

$$
\begin{aligned}
4m_2 \sum_{i \in \mathcal{A}} \frac{p_i(p_i - q_i)^2}{q_i^2} &\leqslant 4m_2 \left( \sum_{i \in \mathcal{A}} \frac{p_i^2}{q_i^2} \right)^{1/2} \left( \sum_{i \in \mathcal{A}} \frac{(p_i - q_i)^4}{q_i^2} \right)^{1/2} \\
&\leqslant 4m_2 \left( 4k + \frac{4}{\alpha m_2} \mathbb{E}[Z_2] \right)^{1/2} \left( \sum_{i \in \mathcal{A}} \frac{(p_i - q_i)^2}{q_i} \right) \\
&= 4 \left( 2\sqrt{k} + 2\sqrt{\frac{1}{\alpha m_2} \mathbb{E}[Z_2]} \right) \mathbb{E}[Z_2] \\
&= 8\sqrt{k}\mathbb{E}[Z_2] + 8(\alpha m_2)^{-1/2}(\mathbb{E}[Z_2])^{3/2}.
\end{aligned}
$$

Combing both, we have that

$$\operatorname{Var}[Z_2] \leqslant 4k + \left( \frac{4}{\alpha m_2} + 8\sqrt{k} \right) \mathbb{E}[Z_2] + 8(\alpha m_2)^{-1/2}(\mathbb{E}[Z_2])^{3/2}. \tag{11}$$

When $d_{\chi^2}(p_{\mathcal{A}}, q_{\mathcal{A}}) \leqslant \varepsilon/2$, then $\mathbb{E}[Z_2] \leqslant \frac{m_2 \varepsilon}{2}$; and we solve $\operatorname{Var}[Z_2] \leqslant (\frac{1}{32} m_2 \varepsilon)^2$, which gives

$$4k + \left( \frac{4}{\alpha m_2} + 8\sqrt{k} \right) \frac{m_2 \varepsilon}{2} + 8(\alpha m_2)^{-1/2}(\frac{m_2 \varepsilon}{2})^{3/2} \leqslant (\frac{1}{32} m_2 \varepsilon)^2.$$

We solve for the relaxation:

$$\text{LHS} \leqslant 4 \cdot \max \left\{ 4k, \frac{2\varepsilon}{\alpha}, 4\sqrt{k} m_2 \varepsilon, 8\frac{m_2}{\sqrt{\alpha} 2^{3/2}} \varepsilon^{3/2} \right\} \leqslant (\frac{1}{32} m_2 \varepsilon)^2$$

In the end, we obtain:

$$\max \left\{ 128 \cdot \frac{\sqrt{k}}{\varepsilon}, 64\sqrt{\frac{2}{\alpha \varepsilon}}, 32^2 \cdot 16\sqrt{k}, 32^2 \cdot 16\frac{1}{\sqrt{\alpha \varepsilon}\sqrt{2}} \right\} \leqslant 32^2 \cdot 16 \cdot \max \left\{ \frac{\sqrt{k}}{\varepsilon}, \sqrt{\frac{1}{\alpha \varepsilon}} \right\} \leqslant m_2$$

When $d_{\chi^2}(p_{\mathcal{A}}, q_{\mathcal{A}}) \geqslant \varepsilon$, then $\mathbb{E}[Z_2] \geqslant m_2 \varepsilon$; and we solve $\operatorname{Var}[Z_2] \leqslant (\frac{1}{4}\mathbb{E}[Z_2])^2$,

$$4k + \left( \frac{4}{\alpha m_2} + 8\sqrt{k} \right) \mathbb{E}[Z_2] + 8(\alpha m_2)^{-1/2}(\mathbb{E}[Z_2])^{3/2} \leqslant (\frac{1}{4}\mathbb{E}[Z_2])^2,$$

which is equivalent to the following

$$\frac{4k}{(\mathbb{E}[Z_2])^{3/2}} + \left( \frac{4}{\alpha m_2} + 8\sqrt{k} \right) \frac{1}{(\mathbb{E}[Z_2])^{1/2}} + 8(\alpha m_2)^{-1/2} \leqslant \frac{1}{16}(\mathbb{E}[Z_2])^{1/2}$$

Further relaxing the solution, it is enough to have

$$\frac{4k}{(\mathbb{E}[Z_2])^{3/2}} + \left(\frac{4}{\alpha m_2} + 8\sqrt{k}\right)\frac{1}{(\mathbb{E}[Z_2])^{1/2}} + 8(\alpha m_2)^{-1/2}$$

$$\leqslant \quad \frac{4k}{(m_2\varepsilon)^{3/2}} + \left(\frac{4}{\alpha m_2} + 8\sqrt{k}\right)\frac{1}{(m_2\varepsilon)^{1/2}} + 8\frac{1}{\sqrt{\alpha m_2}}$$

$$\leqslant \quad \frac{1}{16}(m_2\varepsilon)^{1/2} \leqslant \frac{1}{16}(\mathbb{E}[Z_2])^{1/2},$$

as long as the following holds,

$$m_2 \geqslant 64 \max\left\{\frac{2\sqrt{k}}{\varepsilon}, 2\sqrt{\frac{1}{\alpha\varepsilon}}, 8\frac{\sqrt{k}}{\varepsilon}, 8\sqrt{\frac{\alpha}{\varepsilon}}\right\} = \max\left\{128\sqrt{\frac{1}{\alpha\varepsilon}}, 512\frac{\sqrt{k}}{\varepsilon}\right\}. \qquad (12)$$

Letting $m_2 \geqslant 512 \max\left\{\sqrt{\frac{1}{\alpha\varepsilon}}, \frac{\sqrt{k}}{\varepsilon}\right\}$, we have that both statements. $\qquad\square$

## C   Proofs of entropy testing lower bounds

**Lemma 2.6.** *With fewer than $c_3 \cdot \sqrt{k}/\varepsilon$ samples from $p$, no tester can distinguish between $p = q$ and $|H(p) - H(q)| \geqslant \varepsilon$ with probability higher than $2/3$, where $c_3 > 0$ is an absolute constant.*

*Proof of Lemma 2.6.* This follows from the standard uniformity testing lower bound [Pan08], which provides a lower bound of $\Omega(\sqrt{k}/\eta^2)$: there exists a family of distributions that are hard to distinguish from uniform $u_k$, using fewer than $c_1 \cdot \sqrt{k}/\eta$ samples. Let $k$ be an even number; the construction is by taking $\theta = \{-1, 1\}^{k/2}$ uniformly at random, and letting, for every $i \in [k/2]$,

$$p_\theta^{\text{no}}(2i) = \frac{1 + \theta_i \cdot \eta}{k}, \qquad p_\theta^{\text{no}}(2i + 1) = \frac{1 - \theta_i \cdot \eta}{k}.$$

We can verify that for any $\theta$:

$$|H(p_\theta^{\text{no}}) - H(u_k)| = \log k - \frac{k}{2}\left(\frac{1+\eta}{k}\log\left(\frac{1+\eta}{k}\right) + \frac{1-\eta}{k}\log\left(\frac{1-\eta}{k}\right)\right) = \Theta(\eta^2)$$

Setting $\eta = \varepsilon^2$ yields the lower bound of $\Omega\left(\frac{\sqrt{k}}{\varepsilon}\right)$. $\qquad\square$

**Lemma 2.7.** *With fewer than $c_4 \cdot \log^2 k/\varepsilon^2$ samples from $p$, no tester can distinguish between $p = q$ and $|H(p) - H(q)| \geqslant \varepsilon$ with probability higher than $2/3$, where $c_4 > 0$ is an absolute constant.*

*Proof of Lemma 2.7.* Following [WY16, B.2 Proof of Proposition 2], we look at the same construction but with different parameters $\varepsilon' = \frac{\varepsilon}{\log(2(k-1))}$:

$$p = \left(\frac{1}{3(k-1)}, \ldots, \frac{1}{3(k-1)}, \frac{2}{3}\right), \quad q = \left(\frac{1+\varepsilon'}{3(k-1)}, \ldots, \frac{1+\varepsilon'}{3(k-1)}, \frac{2-\varepsilon'}{3}\right).$$

One can check that

$$H(q) - H(p) \geqslant \frac{1}{3}\log(2(k-1))\varepsilon' - \varepsilon'^2 = \Omega(\varepsilon).$$

Moreover, direct calculation of the (squared) Hellinger distance shows that

$$d_{\mathrm{H}}(p, q)^2 = \Theta(\varepsilon'^2) = \Theta\left(\frac{\varepsilon^2}{\log^2 k}\right)$$

which implies that $p$ and $q$ cannot be distinguished with fewer than $c_4 \frac{\log^2 k}{\varepsilon^2}$ samples [BY02, Theorem 4.7]. $\qquad\square$

# D   Sketch proof of $O\left( \frac{\sqrt{k \log \log \log(k/\varepsilon)}}{\varepsilon} + \frac{\log^2(k)}{\varepsilon^2} \right)$ upper bound.

We will rely the following inequality for compression, both via Jensen's inequality,

$$\left( \sum_{i \in \Delta} p_i \right) \log \left( \frac{1}{\sum_{i \in \Delta} p_i} \right) \leqslant \sum_{i \in \Delta} p_i \log \frac{1}{p_i} \leqslant \left( \sum_{i \in \Delta} p_i \right) \log \left( \frac{|\Delta|}{\sum_{i \in \Delta} p_i} \right), \tag{13}$$

as $\log(x)$ is concave and $\log\left(\frac{1}{x}\right)$ is convex.

$$\sum_{i \in \Delta} p_i \log \frac{1}{p_i} \geqslant \left( \sum_{i \in \Delta} p_i \right) \log \left( \frac{\sum_{i \in \Delta} p_i}{\sum_{i \in \Delta} p_i^2} \right) \geqslant \left( \sum_{i \in \Delta} p_i \right) \log \left( \frac{1}{\sum_{i \in \Delta} p_i} \right),$$

suggesting that if we merge elements of $\Delta$ into one, then we will lose a $\log(|\Delta|)$ factor of the entropy. By merging enough elements, we can then reduce this problem into the first case, where elements have large enough mass in each location.

**Claim D.1.** *Let $\mathcal{S} \subseteq [k]$, if $p(\mathcal{S}) - q(\mathcal{S}) > -\eta$, then*

$$|d_{\mathrm{KL}}(p_{\mathcal{S}}\|q_{\mathcal{S}})| \geqslant \eta \Rightarrow d_{\mathrm{KL}}(p_{\mathcal{S}}\|q_{\mathcal{S}}) \geqslant \eta.$$

*Proof.* If $|d_{\mathrm{KL}}(p_{\mathcal{S}}\|q_{\mathcal{S}})| \geqslant \eta$, then

$$d_{\mathrm{KL}}(p_{\mathcal{S}}\|q_{\mathcal{S}}) \geqslant \eta \text{ or } d_{\mathrm{KL}}(p_{\mathcal{S}}\|q_{\mathcal{S}}) \leqslant -\eta.$$

We will rule out the second case, using that $\log \frac{1}{x} \geqslant 1 - x$ for $x > 0$, [10]

$$d_{\mathrm{KL}}(p_{\mathcal{S}}\|q_{\mathcal{S}}) = \sum_{i \in \mathcal{S}} p_i \log \frac{p_i}{q_i} \geqslant \sum_{i \in \mathcal{S}} p_i \left( 1 - \frac{q_i}{p_i} \right) = p(\mathcal{S}) - q(\mathcal{S}) > -\eta. \tag{14}$$

Thus, we cannot have $d_{\mathrm{KL}}(p_{\mathcal{S}}\|q_{\mathcal{S}}) \leqslant -\eta$ and so $d_{\mathrm{KL}}(p_{\mathcal{S}}\|q_{\mathcal{S}}) \geqslant \eta$. $\qquad\square$

We use the same idea as our first upper bound but choose a series of thresholds. Let

$$\mathcal{S}_3 := \left\{ i \in [k] | q_i \geqslant \Omega\left( \frac{\varepsilon}{k \log \log \log(k/\varepsilon)} \right) \right\};$$

$$\mathcal{S}_2 = \left\{ i \in [k] | \Omega\left( \frac{\varepsilon}{k \log \log(k/\varepsilon)} \right) \leqslant q_i \leqslant O\left( \frac{\varepsilon}{k \log \log \log(k/\varepsilon)} \right) \right\};$$

$$\mathcal{S}_1 = \left\{ i \in [k] | \Omega\left( \frac{\varepsilon}{k \log(k/\varepsilon)} \right) \leqslant q_i \leqslant O\left( \frac{\varepsilon}{k \log \log(k/\varepsilon)} \right) \right\}.$$

The following calculation ensues

$$\begin{aligned}
\Omega(\varepsilon) &\leqslant |H(p_{\mathcal{A}}) - H(q_{\mathcal{A}})| \\
&\leqslant \left| \sum_{i=1}^{3} d_{\mathrm{KL}}(p_{\mathcal{S}_i}, q_{\mathcal{S}_i}) \right| + \left| \sum_{i \in \mathcal{A}} (p_i - q_i) \log \frac{1}{q_i} \right| \\
&\leqslant \sum_{i=1}^{3} |d_{\mathrm{KL}}(p_{\mathcal{S}_i}, q_{\mathcal{S}_i})| + \left| \sum_{i \in \mathcal{A}} (p_i - q_i) \log \frac{1}{q_i} \right|.
\end{aligned}$$

We have that one of the four terms will be at least $\Omega(\varepsilon/4)$. If it is

$$\left| \sum_{i \in \mathcal{A}} (p_i - q_i) \log \frac{1}{q_i} \right| \geqslant \Omega(\varepsilon),$$

which is testable with $O\left( \frac{\log^2(k)}{\varepsilon^2} \right)$ samples using arguments from proof of Theorem 2.1. If it is $|d_{\mathrm{KL}}(p_{\mathcal{S}_i}, q_{\mathcal{S}_i})| \geqslant \Omega(\varepsilon)$, for $i = 1, 2, 3$. We have the following:

---

[10]In the case of $p_i = 0$, we still have $p_i \log\left( \frac{p_i}{q_i} \right) \geqslant p_i - q_i$.

**Case $\mathcal{S}_3$.** Suppose $|d_{\mathrm{KL}}(p_{\mathcal{S}_3}, q_{\mathcal{S}_3})| \geqslant \Omega(\varepsilon)$. We will check whether $p(\mathcal{S}_3) \geqslant \Omega\left(\frac{\varepsilon}{\log\log\log(k/\varepsilon)}\right)$, if not, we can reject. We proceed assuming the inequality holds. Note that

$$|p(\mathcal{S}_3) - q(\mathcal{S}_3)| = |p(\overline{\mathcal{S}_3}) - q(\overline{\mathcal{S}_3})| \leqslant O\left(\frac{\varepsilon}{\log\log\log(k/\varepsilon)}\right).$$

Thus, $p(\overline{\mathcal{S}_3}) - q(\overline{\mathcal{S}_3}) > -O\left(\frac{\varepsilon}{\log\log\log(k/\varepsilon)}\right)$, and by Claim D.1, we have that $d_{\mathrm{KL}}(p_{\mathcal{S}_3}, q_{\mathcal{S}_3}) \geqslant \Omega(\varepsilon)$. Using (4), we then have that $d_{\chi^2}(p_{\mathcal{S}_3}, q_{\mathcal{S}_3}) \geqslant \Omega(\varepsilon)$. Using Lemma 2.4 (setting $\alpha = O\left(\frac{\varepsilon}{k\log\log\log(k/\varepsilon)}\right)$), and similar argument from the proof of Theorem 2.1, we have that $O\left(\frac{\sqrt{k\log\log\log(k/\varepsilon)}}{\varepsilon}\right)$ suffices to check between the case that $d_{\chi^2}(p_{\mathcal{S}_3}, q_{\mathcal{S}_3}) \geqslant \Omega(\varepsilon)$ and $p_{\mathcal{S}_3} = q_{\mathcal{S}_3}$.

**Case $\mathcal{S}_2$.** Suppose $|d_{\mathrm{KL}}(p_{\mathcal{S}_2}, q_{\mathcal{S}_2})| \geqslant \Omega(\varepsilon)$. We will check whether

$$\Omega\left(\frac{\varepsilon}{\log\log(k/\varepsilon)}\right) \leqslant p(\mathcal{S}_2) \leqslant O\left(\frac{\varepsilon}{\log\log\log(k/\varepsilon)}\right),$$

if not, we will reject. We proceed assuming the inequality holds. Now, recall that the main bottleneck of the $\chi^2$ tester analyzed in Lemma 2.4 is due to the minimum probability $\alpha = \min_{i \in \mathcal{S}_2} q_i$ (increasing this would decrease the sample complexity). Our main idea here is to increase $\alpha$ by merging a suitable number ($\log\log(k/\varepsilon)$ in this case) of elements into one single bin to form a new distribution to test. Denote $\Delta_j$ where $j \in \left[\frac{|\mathcal{S}_2|}{\log\log(k/\varepsilon)}\right]$ and $\bigcup_j \Delta_j = \mathcal{S}_2$. We will subsequently treat every elements in $\Delta_j$ as 1 bin in the new distribution, calling it $p_\Delta, q_\Delta$ and denote $p(\Delta_j), q(\Delta_j)$ as mass on $\Delta_j$, where $p(\Delta_j) = \sum_{i \in \Delta_j} p_i$. This gives us the following:

i. $q(\Delta_j) \geqslant \Omega\left(\frac{\varepsilon}{k\log\log(k/\varepsilon)}\right) \cdot |\Delta_j| \geqslant \Omega\left(\frac{\varepsilon}{k}\right)$; the domain size is $\frac{|\mathcal{S}_2|}{\min_j |\Delta_j|} \leqslant O\left(\frac{k}{\log\log(k/\varepsilon)}\right)$.

ii. $\sum_j p(\Delta_j) = p(\mathcal{S}_2) \leqslant O\left(\frac{\varepsilon}{\log\log\log(k/\varepsilon)}\right)$ and $\sum_j q(\Delta_j) = q(\mathcal{S}_2) \leqslant O\left(\frac{\varepsilon}{\log\log\log(k/\varepsilon)}\right)$.

iii. Their entropy difference is preserved, which we will prove next:

$$\left|\sum_j p(\Delta_j) \log\frac{1}{p(\Delta_j)} - \sum_j q(\Delta_j) \log\frac{1}{q(\Delta_j)}\right| \geqslant \Omega(\varepsilon).$$

Note that these are better conditions compared to i. and ii. in the proof of Theorem 2.1 (in this analysis, using ii., it is sufficient to prove that $d_{\mathrm{KL}}(p_\Delta, q_\Delta) \geqslant \Omega(\varepsilon)$ in view of Claim D.1). The gain comes from the fact that we can apply Lemma 2.4 with better $\alpha = \min_j q(\Delta_j) \geqslant \Omega\left(\frac{\varepsilon}{k}\right)$ and thus

$$O\left(\sqrt{\frac{1}{\alpha\varepsilon}} + \sqrt{\frac{k'}{\varepsilon}}\right) = O\left(\sqrt{\frac{k}{\varepsilon^2}}\right) = O\left(\frac{\sqrt{k}}{\varepsilon}\right).$$

However, the gain only affect Claim 2.3 by constant factors. The soundness and completeness then follows similarly to the proof of Theorem 2.1. We prove (iii.) next:

Suppose $H(p_{\mathcal{S}_2}) - H(q_{\mathcal{S}_2}) \geqslant \varepsilon$, then,

$$
\begin{aligned}
\Omega(\varepsilon) & \\
\leqslant \ & \sum_{l \in \mathcal{S}_2} p_l \log \frac{1}{p_l} - \sum_{l \in \mathcal{S}_2} q_l \log \frac{1}{q_l} \\
= \ & \sum_j \sum_{i \in \Delta_j} p_{i,j} \log \frac{1}{p_{i,j}} - \sum_j \sum_{i \in \Delta_j} q_{i,j} \log \frac{1}{q_{i,j}} \\
\leqslant \ & \sum_j p(\Delta_j) \log \frac{|\Delta_j|}{p(\Delta_j)} - \sum_j q(\Delta_j) \log \frac{1}{q(\Delta_j)} \qquad (15) \\
= \ & \sum_j p(\Delta_j) \log |\Delta_j| + \sum_j p(\Delta_j) \log \frac{1}{p(\Delta_j)} - \sum_j q(\Delta_j) \log \frac{1}{q(\Delta_j)}. \\
\leqslant \ & O\left(\frac{\varepsilon}{\log\log\log(k/\varepsilon)}\right) \max_j \log |\Delta_j| + \sum_j p(\Delta_j) \log \frac{1}{p(\Delta_j)} - \sum_j q(\Delta_j) \log \frac{1}{q(\Delta_j)}. (16)
\end{aligned}
$$

where the (15) is due to (13) and for (16), recall that $\sum_j p(\Delta_j) = p(\mathcal{S}_2) \leqslant O\left(\frac{\varepsilon}{\log\log\log(k/\varepsilon)}\right)$.

Suppose $H(q_{\mathcal{S}_2}) - H(p_{\mathcal{S}_2}) \geqslant \Omega(\varepsilon)$, the same goes below:

$$
\begin{aligned}
\Omega(\varepsilon) \ \leqslant \ & \sum_l q_l \log \frac{1}{q_l} - \sum_l p_l \log \frac{1}{p_l} \\
= \ & \sum_j \sum_{i \in \Delta_j} q_{i,j} \log \frac{1}{q_{i,j}} - \sum_j \sum_{i \in \Delta_j} p_{i,j} \log \frac{1}{p_{i,j}} \\
\leqslant \ & \sum_j q(\Delta_j) \log \frac{|\Delta_j|}{q(\Delta_j)} - \sum_j p(\Delta_j) \log \frac{1}{p(\Delta_j)} \\
\leqslant \ & O(\varepsilon) + \sum_j q(\Delta_j) \log \frac{1}{q(\Delta_j)} - \sum_j p(\Delta_j) \log \frac{1}{p(\Delta_j)}.
\end{aligned}
$$

Therefore, we have proved (iii.).

**Case $\mathcal{S}_1$.** The proof follow similar to Case $\mathcal{S}_2$, but by merging $\log(k/\varepsilon)$ elements.

# E  Proofs of testing Bayesian networks

**Claim E.1.** *Suppose that $p(\bar{\mathcal{U}}) \leqslant O(\varepsilon^2/\log(1/\varepsilon))$, then we have for any distribution $q$,*

$$
d_{\mathrm{KL}}(p_{\bar{\mathcal{U}}} \| q_{\bar{\mathcal{U}}}) \geqslant -p(\bar{\mathcal{U}}) \cdot \log\left(\frac{q(\bar{\mathcal{U}})}{p(\bar{\mathcal{U}})}\right) \geqslant -O(\varepsilon^2).
$$

*This implies that $d_{\mathrm{KL}}(p_{\bar{\mathcal{U}}} \| p_{G;\bar{\mathcal{U}}}) \geqslant -O(\varepsilon^2)$ for any DAG G. Moreover, if $d_{\mathrm{KL}}(p \| p_G) \leqslant O(\varepsilon^2)$, then $d_{\mathrm{KL}}(p_{\mathcal{U}} \| p_{G;\mathcal{U}}) \leqslant O(\varepsilon^2)$.*

*Proof of Claim E.1.*

$$d_{\mathrm{KL}}(p_{\bar{\mathcal{U}}}\|p_{G;\bar{\mathcal{U}}}) = \sum_{x\in\bar{\mathcal{U}}} p(x)\log\frac{p(x)}{p_G(x)}$$

$$= -\sum_{x\in\bar{\mathcal{U}}} p(x)\log\frac{p_G(x)}{p(x)}$$

$$\geqslant -\left(\sum_{x\in\bar{\mathcal{U}}} p(x)\right)\cdot\log\left(\frac{\sum_{x\in\bar{\mathcal{U}}} p(x)\cdot\frac{p_G(x)}{p(x)}}{\sum_{x\in\bar{\mathcal{U}}} p(x)}\right)$$

$$= -p(\bar{\mathcal{U}})\cdot\log\left(\frac{p_G(\bar{\mathcal{U}})}{p(\bar{\mathcal{U}})}\right)$$

$$\geqslant -O(\varepsilon^2/\log(1/\varepsilon))\cdot\log\left(\frac{1}{\varepsilon^2/\log(1/\varepsilon)}\right)$$

$$\geqslant -O(\varepsilon^2),$$

where we use monotonicity of $-x\log\frac{1}{x}$ and the fact that $p_G(\bar{\mathcal{U}})\leqslant 1$ in the second last inequality. Since $d_{\mathrm{KL}}(p\|p_G) = d_{\mathrm{KL}}(p_{\mathcal{U}}\|p_{G;\mathcal{U}}) + d_{\mathrm{KL}}(p_{\bar{\mathcal{U}}}\|p_{G;\bar{\mathcal{U}}})\leqslant O(\varepsilon^2)$ and $d_{\mathrm{KL}}(p_{\bar{\mathcal{U}}}\|p_{G;\bar{\mathcal{U}}})\geqslant -O(\varepsilon^2)$, we can rearrange and see that $d_{\mathrm{KL}}(p_{\mathcal{U}}\|p_{G;\mathcal{U}})\leqslant O(\varepsilon^2)$. $\square$

We will need the following claim for the proof of Lemma 3.3.

**Claim E.2.** *The following inequalities hold, for any $i\in[n]$*

$$\sum_{x'\in\mathcal{A}_i'} p_{X_i,\Pi_i^G}(x')\log\frac{p(\pi_i^G(x'))}{q(\pi_i^G(x'))} \geqslant -O\left(\frac{\varepsilon^2}{n^2}\right). \tag{17}$$

$$\sum_{x\in\mathcal{A}_i\setminus\mathcal{U}} p(x)\log\frac{p(x_i(x)|\pi_i^G(x))}{q(x_i(x)|\pi_i^G(x))} \geqslant -O\left(\frac{\varepsilon^2}{n}\right). \tag{18}$$

*Proof.* We will show (17) first.

$$\sum_{x'\in\mathcal{A}_i'} p_{X_i,\Pi_i^G}(x')\log\frac{p(\pi_i^G(x'))}{q(\pi_i^G(x'))} \geqslant \left(\sum_{x'\in\mathcal{A}_i'} p_{X_i,\Pi_i^G}(x')\right)\cdot\log\left(\frac{\sum_{x'\in\mathcal{A}_i'} p_{X_i,\Pi_i^G}(x')}{\sum_{x'\in\mathcal{A}_i'} p_{X_i,\Pi_i^G}(x')\cdot\frac{q(\pi_i^G(x'))}{p(\pi_i^G(x'))}}\right)$$

$$\geqslant \left(\sum_{x'\in\mathcal{A}_i'} p_{X_i,\Pi_i^G}(x')\right)\cdot\log\left(\frac{\sum_{x'\in\mathcal{A}_i'} p_{X_i,\Pi_i^G}(x')}{\sum_{(x,\pi)\in\mathcal{A}_i'} p_{X_i|\Pi_i^G}(x|\pi)\cdot q(\pi)}\right)$$

$$\geqslant \left(\sum_{x'\in\mathcal{A}_i'} p_{X_i,\Pi_i^G}(x')\right)\cdot\log\left(\sum_{x'\in\mathcal{A}_i'} p_{X_i,\Pi_i^G}(x')\right)$$

$$\geqslant \left(1-O\left(\frac{\varepsilon^2}{n^2\log(n/\varepsilon)}\right)\right)\log\left(1-O\left(\frac{\varepsilon^2}{n^2\log(n/\varepsilon)}\right)\right)$$

$$\geqslant -O\left(\frac{\varepsilon^2}{n^2\log(n/\varepsilon)}\right)$$

where the first step follows from Jensen's inequality applied to the function $f(x) = \log(1/x)$, and the second-to-last step is due to $x\log x$ is monotonically increasing when $x\geqslant\frac{1}{e}$ and $\log(1-x)\geqslant -2x$ when $x\in(0,0.5)$. This concludes the proof of (17).

We next move on to proving (18).

$$\sum_{x \in \mathcal{A}_i \setminus \mathcal{U}} p(x) \log \frac{p(x_i(x)|\pi_i^G(x))}{q(x_i(x)|\pi_i^G(x))} \geqslant \left(\sum_{x \in \mathcal{A}_i \setminus \mathcal{U}} p(x)\right) \log \left(\frac{\sum_{x \in \mathcal{A}_i \setminus \mathcal{U}} p(x)}{\sum_{x \in \mathcal{A}_i \setminus \mathcal{U}} p(x) \cdot \frac{q(x_i(x)|\pi_i^G(x))}{p(x_i(x)|\pi_i^G(x))}}\right)$$

$$= \left(\sum_{x \in \mathcal{A}_i \setminus \mathcal{U}} p(x)\right) \log \left(\frac{\sum_{x \in \mathcal{A}_i \setminus \mathcal{U}} p(x)}{\sum_{x \in \mathcal{A}_i \setminus \mathcal{U}} \frac{p(x)}{p(x_i(x)|\pi_i^G(x))} q(x_i(x)|\pi_i^G(x))}\right)$$

$$\geqslant \left(\sum_{x \in \mathcal{A}_i \setminus \mathcal{U}} p(x)\right) \log \left(\sum_{x \in \mathcal{A}_i \setminus \mathcal{U}} p(x)\right)$$

$$\geqslant -O\left(\frac{\varepsilon^2}{n}\right),$$

where the second-to-last step follows from Equation (19) below, and last step by monotonicity (decreasing) of $f(x) = x \log x$ when $x \leqslant \frac{1}{e}$ and $\sum_{x \in \mathcal{A}_i \setminus \mathcal{U}} p(x) \leqslant \sum_{i \neq j} \sum_{x \in \mathcal{A}_j} p(x) \leqslant \sum_{i \neq j} O\left(\frac{\varepsilon^2}{n^2 \log(n/\varepsilon)}\right) \leqslant O\left(\frac{\varepsilon^2}{n \log(n/\varepsilon)}\right).$

$$\sum_{x \in \mathcal{A}_i \setminus \mathcal{U}} \frac{p(x)}{p(x_i(x)|\pi_i^G(x))} q(x_i(x)|\pi_i^G(x))$$

$$= \sum_{x \in \mathcal{A}_i \setminus \mathcal{U}} \frac{p(x)}{p(x_i(x)|\pi_i^G(x))} q(x_i(x)|\pi_i^G(x))$$

$$= \sum_{x \in \mathcal{A}_i \setminus \mathcal{U}} \frac{p(\pi_i^G(x))q(x_i(x)|\pi_i^G(x))}{p(x_i(x), \pi_i^G(x))} p(x)$$

$$= \sum_{x \in \mathcal{A}_i \setminus \mathcal{U}} p(\pi_i^G(x))q(x_i(x)|\pi_i^G(x)) \cdot p(x|x_i(x), \pi_i^G(x))$$

$$\leqslant \sum_{(x_i, \pi_i^G) \in \{0,1\}^{|\Pi_i^G|+1}} p(\pi_i^G)q(x_i|\pi_i^G) \left(\underbrace{\sum_{x' \in \{0,1\}^{n-|\Pi_i|+1}} p(x'|x_i, \pi_i^G)}_{=1}\right) \qquad (19)$$

$$= \sum_{\pi_i^G \in \{0,1\}^{|\Pi_i^G|}} p(\pi_i^G) \underbrace{\sum_{x_i \in \{0,1\}} q(x_i|\pi_i^G)}_{=1} = 1.$$

This concludes the proof of (18). $\qquad \square$

**Lemma 3.3.** *Suppose* $d_H^2(p,q) \geqslant \Omega(\varepsilon^2)$; $d_{KL}(p \| p_G) \leqslant O(\varepsilon^2)$; $p(\bar{\mathcal{U}}) \leqslant \frac{\varepsilon^2}{n \log(n/\varepsilon)}$; $\forall i \in [n], p(\bar{\mathcal{A}}_i') \leqslant \frac{\varepsilon^2}{n^2 \log(n/\varepsilon)}$, *where* $\mathcal{A}_i'$ *is defined above, and* $q$ *is Markov with respect to* $G$, *then we have*

$$\sum_{i=1}^n d_{KL}(p_{X_i, \Pi_i^G; \mathcal{A}_i'} \| q_{X_i, \Pi_i^G; \mathcal{A}_i'}) \geqslant \Omega(\varepsilon^2).$$

*Therefore testing* $d_{KL}(p_{X_i, \Pi_i^G; \mathcal{A}_i'} \| q_{X_i, \Pi_i^G; \mathcal{A}_i'}) \geqslant \frac{\varepsilon^2}{n}$ *over all* $i$ *suffices to detect this case.*

*Proof of Lemma 3.3.* By Claim E.1 and the assumption that $d_{KL}(p \| p_G) \leqslant O(\varepsilon^2)$, we have that

$$d_{KL}(p_{\mathcal{U}} \| p_{G;\mathcal{U}}) = d_{KL}(p \| p_G) - d_{KL}(p_{\bar{\mathcal{U}}} \| p_{G;\bar{\mathcal{U}}}) \leqslant O(\varepsilon^2). \qquad (20)$$

$$\Omega(\varepsilon^2) \leqslant d_H^2(p,q) = d_H^2(p_{\mathcal{U}}, q_{\mathcal{U}}) + d_H^2(p_{\bar{\mathcal{U}}}, q_{\bar{\mathcal{U}}}) \leqslant d_H^2(p_{\mathcal{U}}, q_{\mathcal{U}}) + \frac{1}{2}(p(\bar{\mathcal{U}}) + q(\bar{\mathcal{U}})) \leqslant d_H^2(p_{\mathcal{U}}, q_{\mathcal{U}}) + O(\varepsilon^2).$$

$$\Omega(\varepsilon^2) \leqslant d_H^2(p_\mathcal{U}, q_\mathcal{U}) \leqslant d_{\mathrm{KL}}(p_\mathcal{U}\|q_\mathcal{U}) + q(\mathcal{U}) - p(\mathcal{U}) \Rightarrow d_{\mathrm{KL}}(p_\mathcal{U}\|q_\mathcal{U}) \geqslant \Omega(\varepsilon^2). \qquad (21)$$

Combining and (20) and (21), we write

$$
\begin{aligned}
\Omega(\varepsilon^2) \quad \leqslant \quad & d_{\mathrm{KL}}(p_\mathcal{U}\|q_\mathcal{U}) - d_{\mathrm{KL}}(p_\mathcal{U}\|p_{G;\mathcal{U}}) \\[1mm]
= \quad & \sum_{x \in \mathcal{U}} p(x) \sum_{i=1}^n \log \frac{p(x_i(x)|\pi_i^G(x))}{q(x_i(x)|\pi_i^G(x))} \\[1mm]
= \quad & \sum_{i=1}^n \left( \sum_{x \in \mathcal{A}_i} p(x) \log \frac{p(x_i(x)|\pi_i^G(x))}{q(x_i(x)|\pi_i^G(x))} - \sum_{x \in \mathcal{A}_i \setminus \mathcal{U}} p(x) \log \frac{p(x_i(x)|\pi_i^G(x))}{q(x_i(x)|\pi_i^G(x))} \right) \\[1mm]
= \quad & \left( \sum_{i=1}^n \sum_{x \in \mathcal{A}_i} p(x) \log \frac{p(x_i(x)|\pi_i^G(x))}{q(x_i(x)|\pi_i^G(x))} \right) - \left( \sum_{i=1}^n \sum_{x \in \mathcal{A}_i \setminus \mathcal{U}} p(x) \log \frac{p(x_i(x)|\pi_i^G(x))}{q(x_i(x)|\pi_i^G(x))} \right) \\[1mm]
= \quad & \left( \sum_{i=1}^n \sum_{x' \in \mathcal{A}_i'} p_{X_i, \Pi_i^G}(x') \log \frac{p(x_i(x')|\pi_i^G(x'))}{q(x_i(x')|\pi_i^G(x'))} \right) - \left( \sum_{i=1}^n \sum_{x \in \mathcal{A}_i \setminus \mathcal{U}} p(x) \log \frac{p(x_i(x)|\pi_i^G(x))}{q(x_i(x)|\pi_i^G(x))} \right) \\[1mm]
= \quad & \sum_{i=1}^n d_{\mathrm{KL}}(p_{X_i, \Pi_i^G; \mathcal{A}_i'}\|q_{X_i, \Pi_i^G; \mathcal{A}_i'}) - \sum_{i=1}^n \sum_{x' \in \mathcal{A}_i'} p_{X_i, \Pi_i^G}(x') \log \frac{p(\pi_i^G(x'))}{q(\pi_i^G(x'))} \\[1mm]
& - \left( \sum_{i=1}^n \sum_{x \in \mathcal{A}_i \setminus \mathcal{U}} p(x) \log \frac{p(x_i(x)|\pi_i^G(x))}{q(x_i(x)|\pi_i^G(x))} \right)
\end{aligned}
$$

With Claim E.2 and Claim E.1, we can lower bound the sum of local KL divergences,

$$\sum_{i=1}^n d_{\mathrm{KL}}(p_{X_i, \Pi_i^G; \mathcal{A}_i'}\|q_{X_i, \Pi_i^G; \mathcal{A}_i'}) \geqslant \Omega(\varepsilon^2).$$

$\square$

**Lemma 3.4.** *Let $p$ and $q$ be two max in-degree-$d$ Bayes nets supported on $\{0,1\}^n$ such that for every subset $L \subseteq \{X_1, \cdots, X_n\}$ of size $d+1$, the following holds:*

$$|H(p_L) - H(q_L)| \leqslant O\left(\frac{\varepsilon^2}{n}\right).$$

*Suppose $p$ is Markov w.r.t. $G'$ and $q$ Markov w.r.t. $G$. Then we have that*

$$d_{\mathrm{KL}}(p\|p_G) \leqslant O(\varepsilon^2) \text{ and } d_{\mathrm{KL}}(q\|q_{G'}) \leqslant O(\varepsilon^2).$$

*Proof of Lemma 3.4.* More formally, by the celebrated works of Chow and Liu [CL68] and its generalization to Bayes net, one can write the KL divergence between a Bayes net and its projection to any graph $G$ as difference between sum of $n$ local conditional entropies (we provide a derivation for the sake of completeness in Appendix F):

$$0 \leqslant d_{\mathrm{KL}}(p\|p_G) = -\sum_{i=1}^n H(p_{X_i, X_{\Pi_i}}|p_{X_{\Pi_i}}) + \sum_{i=1}^n H(p_{X_i, X_{\Pi_i^G}}|p_{X_{\Pi_i^G}}), \qquad (22)$$

$$0 \leqslant d_{\mathrm{KL}}(q\|q_{G'}) = -\sum_{i=1}^n H(q_{X_i, X_{\Pi_i^G}}|q_{X_{\Pi_i^G}}) + \sum_{i=1}^n H(q_{X_i, X_{\Pi_i}}|q_{X_{\Pi_i}}), \qquad (23)$$

where $X_{\Pi_i}$ denotes the parents of $X_i$ in Bayes net $p$ and $X_{\Pi_i^G}$ denotes the parents of $X_i$ in DAG $G$. Here we assume that $q$ is Markov with respect to $G$. Since the local entropies between $p$ and $q$ are close by $O(\varepsilon^2/n)$ (see (6)) and its relation to conditional entropy via (3), we can conclude the following:

$$H(q_{X_i, X_{\Pi_i}}|q_{X_{\Pi_i}}) - O(\varepsilon^2/n) \leqslant H(p_{X_i, X_{\Pi_i}}|p_{X_{\Pi_i}}) \leqslant H(q_{X_i, X_{\Pi_i}}|q_{X_{\Pi_i}}) + O(\varepsilon^2/n). \qquad (24)$$

$$H(q_{X_i, X_{\Pi_i^G}} | q_{X_{\Pi_i^G}}) - O(\varepsilon^2/n) \leqslant H(p_{X_i, X_{\Pi_i^G}} | p_{X_{\Pi_i^G}}) \leqslant H(q_{X_i, X_{\Pi_i^G}} | q_{X_{\Pi_i^G}}) + O(\varepsilon^2/n). \quad (25)$$

Since we assume that $q$ is Markov with respect to $G$, we can combine (22), (23), (24) and (25), which then give:

$$d_{\mathrm{KL}}(p\|p_G) \leqslant - \sum_{i=1}^n H(q_{X_i, X_{\Pi_i}} | q_{X_{\Pi_i}}) + \sum_{i=1}^n H(q_{X_i, X_{\Pi_i^G}} | q_{X_{\Pi_i^G}}) + O(\varepsilon^2) = -d_{\mathrm{KL}}(q\|q_{G'}) + O(\varepsilon^2),$$

where $p$ is Markov with respect to $G'$, as $\Pi_i$ is the set of parents of $X_i$ for $p$. Rearranging terms and we have

$$d_{\mathrm{KL}}(p\|p_G) + d_{\mathrm{KL}}(q\|q_{G'}) \leqslant O(\varepsilon^2). \quad (26)$$

Since KL-divergence is nonnegative, we conclude that both terms must be at most $O(\varepsilon^2)$. $\qquad \square$

## F   Derivation for KL decomposition

Below, we provide a full proof on decomposing the KL divergence between a Bayes net $p$ and its projection $p_G$, for any DAG $G$, into local conditional entropies.

$$d_{\mathrm{KL}}(p, p_G)$$

$$= \sum_{x \in \{0,1\}^n} p(x) \log \frac{p(x)}{p_G(x)}$$

$$= \sum_{x \in \{0,1\}^n} p(x) \log \frac{\prod_{i=1}^n p(x_i|\pi_i)}{\prod_{i=1}^n p_G(x_i|\pi_i^G)}$$

$$= \sum_{x \in \{0,1\}^n} p(x) \log \left( \prod_{i=1}^n p(x_i|\pi_i) \right) - p(x) \log \left( \prod_{i=1}^n p_G(x_i|\pi_i^G) \right)$$

$$= \sum_{x \in \{0,1\}^n} \sum_{i=1}^n p(x) \log(p(x_i|\pi_i)) - p(x) \log(p_G(x_i|\pi_i^G))$$

$$= \sum_{i=1}^n \sum_{x \in \{0,1\}^n} p(x) \log(p(x_i|\pi_i)) - p(x) \log(p_G(x_i|\pi_i^G))$$

$$= \sum_{i=1}^n \left( \sum_{x_i, \pi_i \in X_i, \Pi_i} p(x_i, \pi_i) \log(p(x_i|\pi_i)) \right) - \left( \sum_{x_i, \pi_i \in X_i, \Pi_i^G} p(x_i, \pi_i^G) \log(p_G(x_i|\pi_i^G)) \right)$$

$$= \sum_{i=1}^n H(p_{X_i, \Pi_i^G} | p_{\Pi_i^G}) - H(p_{X_i, \Pi_i} | p_{\Pi_i}),$$

where $\pi_i, \Pi_i$ denote the parents of $x_i, X_i$ in Bayes net $p$ (a set of random variables or their domain); and $\pi_i^G, \Pi_i^G$ as the parents defined by $G$. $p_G$ is the projection of $p$ unto $G$ as defined by Definition 3.2. It is not hard to see that the derivation extends beyond the case of hypercube, $\{0,1\}^n$.

