# OpenReview forum: "Entropy testing and its application to testing Bayesian networks"
_NeurIPS.cc/2024/Conference — NeurIPS 2024 poster_

### Official Review · Reviewer_wSfS · 2024-06-24

**Soundness:** 2
**Presentation:** 3
**Contribution:** 3
**Rating:** 5
**Confidence:** 4

**Summary:**

In this paper, the authors first find the complexity upper and lower bound for the entropy identity testing problem: given sample access to a distribution $p$ and a fully described distribution $q$, the tester needs do distinguish between two hypotheses $p=q$ and $|H(p)-H)(q)|\geq \epsilon$. Based on this, the authors find the sample complexity upper bound for the problem of identity testing for in-degree-$d$ $n$-dimensional Bayesian networks. This bound improves an existing bound by Canonne et al. (2020) without using an additional assumption that the structure of unknown Bayes net is a subset of that of the reference one.

**Strengths:**

The claimed results sound interesting.

**Weaknesses:**

Some key proofs look not correct, and some notations are not defined. Please see the questions below.

**Questions:**

+ What is the definition of $\mbox{Poi}(m)$ in Claim 2.3?
+ Proof of Theorem 2.1: The first inequality between lines 195 and 196 looks not correct since the function $\log \frac{1}{x}$ is convex. An application of the Jensen's inequality should give $H(q_{\bar{A}}) \geq q(\bar{A}) \log \frac{|\bar{A}|}{q(\bar{A})}$ (not $\leq$ as in your paper).
+ For the case $p=q$, from the line 206, we only infer that $Z_2<2m_2 \epsilon$ with probability $>3/4$. What happens in Algorithm 1 if $2m_2 \epsilon> Z_2\geq \frac{1}{16}m_2 \epsilon$?
+  In the last equations in Appendix A (Derivation of Line 4 in Algorithm 1), you mentioned that you use Chernoff bound $\mathbf{P}(\hat{p}'(\bar{A})\leq (1-1/3)p(\bar{A}))\leq \exp\big(-\frac{m_1 p(\bar{A})}{18}\big)$. Would you please explain this more? It seems to me that you need use the union bound before using the Chernoff bound, hence we need to multiply $|\bar{A}|$ in the RHS of the above inequality.
+ In the statement of Lemma 2.4, $\mbox{Var}[Z_2]\leq O(m_2^2 \epsilon^2)$ and $\mbox{Var}[Z_2] \leq O(\mathbf{E}[Z_2]^2)$. However, in the proof (from line 204 to line 223), you use the exact constants.
+ In line 218, you mention that you use Claim 2.3 to derive some results. How do you justify that $d_{\chi^2}(p_A,q_A)\leq \epsilon/8$ to apply this claim?

**Limitations:**

This is a theoretical research paper, hence the negative society impact of this work is not direct.

---

> ### Author Rebuttal · Authors · 2024-08-07
>
> Thank you for your time and review (and for taking a close look at the
> proofs).
>
> We would like to address the questions raised in the review below:
>
> 1. As is standard in distribution testing, we work in the Poissonized sampling
> model. $\operatorname{Poi} (m)$ denotes a random variable distributed as the Poisson
> distribution with parameter $m$. In the revision, we will formally define this
> in the Preliminaries.
>
> 2. We apologize for skipping some proof details when applying Jensen's
> inequality here. Our proof means to use the function $f (x) = \log (x)$ as
> concave. So, in this case, we are taking $x_i = \frac{1}{q_i}$ and $a_i = q_i$
> which yields the following inequality:
> $$ \frac{\sum a_i \log (x_i)}{\sum a_i} \leqslant \log \left( \frac{\sum a_i
>    x_i}{\sum a_i} \right) \Rightarrow \frac{\sum q_i \log \left( \frac{1}{q_i}
>    \right)}{\sum q_i} \leqslant \log \left( \frac{\sum q_i
>    \frac{1}{q_i}}{\sum q_i} \right) . $$
> 3. Thank you for catching this! We neglected to change out the constants while
> calculating this part of the proof. Note that, this does not affect the end
> statement of the theorem, but just the constant factors in the analysis.
> Indeed, $\operatorname{Var} [Z_2]$ needs to be smaller in terms of its constant in the
> case of Completeness for the analysis to work; and one can do that by
> tightening the constant in the analysis. In particular, we can adjust from the
> inequality from line 431-433 from the proof of Lemma 2.4. See calculation
> details below:
>
> Solve $2 \sqrt{\operatorname{Var} [Z_2]} \leqslant \frac{1}{16} m_2 \varepsilon$:
> $$ 4 k + \left( \frac{4}{\alpha m_2} + 8 \sqrt{k} \right)  \frac{m_2
>    \varepsilon}{2} + 8 (\alpha m_2)^{- 1 / 2}  \left( \frac{m_2
>    \varepsilon}{2} \right)^{3 / 2} \leqslant \frac{1}{32^2} m_2^2
>    \varepsilon^2 ; $$
> $$ m_2 \geqslant 128 \max \left ( \frac{2 \sqrt{k}}{\varepsilon},
>    \sqrt{\frac{2}{\alpha \varepsilon}}, 4 \frac{\sqrt{k}}{\varepsilon}, 2
>    \sqrt{\frac{2}{\alpha \varepsilon}} \right ) = 512 \max \left(
>    \sqrt{\frac{1}{\alpha \varepsilon}}, \frac{\sqrt{k}}{\varepsilon} \right)
>    . $$
>
> Note that, this is the same requirement as the other case (no change to the
> sample complexity upper bound in terms of constant).
>
> 4. Note that $m_1  \hat{p}' (\bar{A})$ is distributed as Binomial with
> parameter $m_1$, $p (\bar{A})$ and so we can apply Chernoff bound on this
> random variable.
>
> 5. Sorry for the slight abuse of notation. We will add the constants in the
> statement of Lemma 2.4. Specifically, relating this to reponse 3 above: we
> will change the statement to
>
> *Let $\mathcal{A} \leftarrow \{i \in [k] \mid q_i \geqslant \alpha\}$. Let $m_2 \geqslant 512 \max \left(\sqrt{\frac{1}{\alpha \varepsilon}}, \frac{\sqrt{k}}{\varepsilon}\right)$ be the number of samples used to
> compute $Z_2$. Then $\mathbb{E} [Z_2] = m_2 d_{\chi^2} (p_{\mathcal{A}}, q_{\mathcal{A}})$. Moreover, if $d_{\chi^2} (p_{\mathcal{A}}, q_{\mathcal{A}}) \leqslant \frac{\varepsilon}{2}$, then $\operatorname{Var} [Z_2] \leqslant \left( \frac{1}{32} m_2 \varepsilon \right)^2$. If $d_{\chi^2} (p_{\mathcal{A}},q_{\mathcal{A}}) \geqslant \varepsilon$, then $\operatorname{Var} [Z_2] \leqslant \left( \frac{1}{4} \mathbb{E}[Z_2] \right)^2$.*
>
> 6. Sorry for not being clear enough: indeed, it relies on the testing outcome
> from line 215-217, which is saying, if $d_{\chi^2} (p_{\mathcal{A}},
> q_{\mathcal{A}}) \geqslant \frac{1}{8} \varepsilon$, then we can detect it and
> reject it early (with high probability). So to continue towards line 218, and
> still not being rejected, it has to be the case that $d_{\chi^2}
> (p_{\mathcal{A}}, q_{\mathcal{A}}) \leqslant \frac{1}{8} \varepsilon$.

---

> > ### Author Response · Authors · 2024-08-14
> >
> > Getting back to this, we would like to know if the reviewer is satisfied with our answers addressing the correctness doubts they raised; if so, whether they would reflect this in their score, which currently indicates "a paper with technical flaws".

---

### Official Review · Reviewer_S8s6 · 2024-07-11

**Soundness:** 3
**Presentation:** 2
**Contribution:** 3
**Rating:** 5
**Confidence:** 3

**Summary:**

The authors consider the entropy identity testing problem for two discrete distributions $p$ and $q$, which is a hypothesis test between $p=q$ and $|H(p) - H(q)| \geq \epsilon$. They propose an algorithm for this problem that is near-optimal in terms of the sample complexity. The main ideas of the algorithm are 1) ignoring a set with small probability with respect to $q$ and 2) bounding the entropy difference by KL-divergence. The result is applied to identity testing problem for Bayes nets.

**Strengths:**

The performance of the algorithms, including the runtime and the error bound, is guaranteed by theoretical analysis.
Detailed mathematical proof is given.

**Weaknesses:**

The importance of the problem is not discussed much.
The writing is not clear with several errors.

**Questions:**

1) It is hard to see the significance of the entropy identity testing.
Heuristically, the alternative hypothesis $|H(p) - H(q)| > \epsilon$ is too much different from the null, since it is possible that $p \neq q$ but $H(p) =H(q)$,
and thus the entropy identity testing might be essentially easier than the other testing problems based on "distances" such as KL-divergence or Hellinger distance.

2) I think the manuscript should be revised for clarity, since it contains too many typos and small errors. Below are a few examples.
- In Abstract, $k$ is not defined.
- In line 131, what is $\leftarrow$? Also, if $\bar{A}$ is the complement of $A$, then the equality should be removed from $q_i \leq \tau/k$.
- In the equation below line 134, $p(\bar{A}) = O(\tau)$ does not imply the second inequality.
- In line 182, the semicolon seems to be a typo.
- In Algorithm 1, what is $N_i$?
- In line 200 and line 202, "Algorithm 4" should be "Algorithm 1".
- In Algorithm 2, what is S_1?

**Limitations:**

The work does not seem to have potential negative societal impact.

---

> ### Author Rebuttal · Authors · 2024-08-07
>
> Thank you for your time and review.
>
> 1. Indeed, the formulation introduced is a bit atypical, but we emphasize that
> even more standard formulations where the alternative is in terms of distances
> (such as total variation distance or KL divergence, say) might still have a
> similar objection, namely that  $p \neq q$ but $d(p,q) < \varepsilon$. The motivation for
> the formulation chosen is made apparent by one of the applications, namely
> that of Bayesian network testing, where we show that as a building block, it
> leads to savings compared to existing algorithms (this application was
> actually the starting point for formulating this hypothesis testing question
> in the first place). We strongly believe that this new hypothesis testing
> question will find other applications.
>
> 2. Thank you for the comments. We will fix the typos the reviewer pointed out
> and improve the presentation; in particular, defining the notation
> $\leftarrow$ for variable assignment and $N_i$ for the empirical count among
> samples of the $i$-th element; $\mathcal{S}_1$ was the set of samples from
> Line 1 of Algorithm 2, and should have been written there. $p (\bar{A}) = O
> (\tau)$ for sufficiently small constant inside the big $O$, implies the second
> inequality through the monotonicity of $f (x) = x \log \frac{1}{x}$ when $x <
> \frac{1}{e}$.

---

> > ### Comment · Reviewer_S8s6 · 2024-08-10
> >
> > Thank you for your answer. I might rephrase my first question as follows. If you use a 'true' distance then the null $p=q$ is equivalent to $d(p, q)=0$, hence the testing is about whether $d(p, q)=0$ vs. $d(p, q)>\varepsilon$, which is a usual hypothesis test that concerns a single parameter $d(p, q)$. However, in this work, the entropy identity testing is not about whether $|H(p) - H(q)| = 0$ vs. $|H(p) - H(q)| >\varepsilon$, since $p=q$ is essentially different from $|H(p) - H(q)| = 0$. Anyhow, I can see your point that the main motivation of the work was to consider Bayesian networks.

---

> > > ### Author Response · Authors · 2024-08-14
> > >
> > > Thanks for clarifying! Indeed, our null hypothesis is more stringent (smaller set) than the one you mention -- first, in order to have a non-degenerate null hypothesis, but more importantly as this is the "cleanest" formulation for our application. However, relaxing the null to $H(p)=H(q)$ would make the sample complexity of the problem much higher, basically as hard as entropy estimation:
> > > $$
> > > \Theta\left(\frac{k}{\varepsilon\log k} + \frac{\log^2 k}{\varepsilon^2}\right)
> > > $$
> > > (This is because the lower bounds from Wu and Yang (2016) still apply via a reduction, while their upper bound does allow to solve the problem.)

---

### Official Review · Reviewer_cFXh · 2024-07-12

**Soundness:** 4
**Presentation:** 3
**Contribution:** 3
**Rating:** 6
**Confidence:** 4

**Summary:**

* This work focuses on the problem of Entropy identity testing, i.e., whether for two distributions $p, q$ if $p = q$ or $|H(p) - H(q)| > \epsilon $ given samples from unknown  $p$ and complete description of $q$ over a domain of size $k$.

* The authors show that the sample complexity of entropy identity testing is $ \sqrt{k \log(k/\epsilon )} /\epsilon + \log^2(k)/\epsilon^2 $. This is an interesting observation since for identity testing problems, the sample complexity is usually $O(\sqrt{k}/\epsilon^2)$, but now it gets split into two terms with $\epsilon^2$ appearing below the shorter term when $k$ is exponentially large.

* Using this testing procedure as a subroutine, this results leads to better testing procedure for the problem of Identity testing for Bayesian networks over {0,1}$^n$ and the authors (as mentioned in abstract) obtain a better bound for this problem improving it from $O(2^{d/2} n^2 /\epsilon^4 )$ in [CDKS20] to $O(2^{d/2} n \epsilon^2 + n^2/\epsilon^4)$.

Roughly speaking, the algorithm for entropy identity testing divides the space into two regions:
1) Where $q$ has low probability: just test whether $\hat{p}$ attains large values in this region. if so, then reject.
2) Where $q$ has high probability: use the algorithm of [DKW18] to efficiently test whether $d_{KL}(p||q) \geq \epsilon$.

**Strengths:**

* The nice thing is the observation that the identity entropy testing problem is much easier than the entropy estimation problem itself, which is known to be $\theta( \frac{k}{\epsilon \log k}  + \frac{\log^2 k}{\epsilon^2} )$.

* The paper is well explained. I appreciate the intuitive explanation and thought process of bounds that are obtained using simple ideas, followed by the need to make appropriate modifications to overcome those issues.

* It is indeed surprising that entropy, which is not an easy quantity to estimate (optimally), is much easier to test.

**Weaknesses:**

* The algorithm/testing-procedure, although interesting from a theoretical viewpoint, is hardly practical as it involves so many individual tests and just too many weird (but tunable) constants. A "neat algorithm" would have been even more useful from a practical viewpoint, but overall the work is interesting and may serve as a nice starting point for designing a ``neat" algorithm.

Other minor points:
* The (possible) applications of either of the two testing problems, especially the identity testing of degree-d Bayes can be mentioned.

* Lines 142-143 can be slightly rephrased ``directly extending this argument does not work" causes some confusion.

* In line 145, please provide a reference/explanation for where the factor $\epsilon / \sqrt{n}$ comes from. If I am not wrong, it probably comes from [DP16 Theorem 4.2]

**Questions:**

As far as the goal of getting a better bound for identity testing for Bayes net is concerned, won't Reyni Entropy identity testing might be a better choice (especially of order $2$), which has $\sqrt{k}$ sample complexity and is also much simpler to estimate and might have even better sample complexity for identity testing.

**Limitations:**

Work has essentially no negative societal impact. A justification for the limitations question in the checklist can be added as it could help the readers directly without searching at different places.

---

> ### Author Rebuttal · Authors · 2024-08-07
>
> We thank the reviewer for their encouraging comments and feedback.
>
> We did not try to optimize the constants. This is primarily a theoretical
> contribution which can serve as a proof of concept: any practical
> implementation would be significantly optimized, and there is nothing a priori
> inherent to our approach that would make this impossible.
>
> Regarding the comment about Renyi entropy: this is an interesting question.
> Renyi entropy of order 2 is equivalent to $\ell_2$ norm of the distribution,
> and so, in some sense, in testing in $\ell_2$ (which has been considered in
> previous work, but did not appear to yield any obvious sample complexity
> improvement for the task of Bayes net testing). It is not clear to us whether
> this would translate to an algorithm, but this is something to consider in
> future work (especially for general $\alpha$-Renyi entropy).
>
> *On the minor points*:
> 1. 2. Thank you for the suggestions.
> 3.. Yes, you are correct.
>
> We will make the corresponding changes in the revision.

---

> > ### Comment · Reviewer_cFXh · 2024-08-12
> >
> > Thanks for the response. As my questions have been adequately addressed, and I did not identify any major flaws or concerns with the approach, I maintain my current rating, and I believe the paper meets the necessary standards for acceptance.

---

### Official Review · Reviewer_BQBQ · 2024-07-12

**Soundness:** 4
**Presentation:** 3
**Contribution:** 3
**Rating:** 6
**Confidence:** 3

**Summary:**

This paper studies the problem of testing a null hypothesis against alternatives that are far in Shannon entropy. They provide information-theoretic minimax lower bounds that they also achieve up to polylogarithmic factors. They then apply these results to Bayes net testing.

**Strengths:**

* Clear and well-written
* This seems like a fundamental problem and it’s good to get tight bounds for it

**Weaknesses:**

* The lemmas in section 2.2 are written somewhat informally, but they read as if they are making an instance-dependent claim (e.g., for any $q$, it is impossible to test with fewer than $c_3 \sqrt{k}/\varepsilon$). However, the lemmas seem to actually be minimax claims, as the two lower bounds are obtained by different single hard distributions.
* There is no discussion of proof strategy or intuition in section 2.2 so the results feel like they come from no where (the results of lemma 2.6 were discussed in section 1.2, but not lemma 2.7 as far as I can tell).
* The fact that the two lower bounds arise from such different distributions indicates the possibility of a tighter instance-dependent quantity. This sort of bound could help illuminate the structure of this testing problem.

**Questions:**

* In the display after line 195, I don’t quite follow how the last inequality holds. In particular, where does the $\log \log (k/\varepsilon)$ factor go?
* Between equations (13) and (14) in the display following line 416, $d_{TV}(p, q)$ is bounded above by $\sqrt{\frac{\varepsilon}{8}} + 4\tau$, which is then bounded above by a constant. Isn’t it easier to just directly (and trivially) bound $d_{TV}(p, q)$ by 1? This would also remove the requirement (as far as I can tell) of assuming $p(\bar{A}) + q(\bar{A}) \leq 4\tau$.

Minor comments
* In line 108, in the phrase “as just mentioned, when its TV distance…” what is “it” referring to?
* In line 163, is the partial Hellinger distance missing a power of 2 in the summand?
* In lines 414-416, the sentence “We continue based on the premise…” is redundant, as the premises given are already assumed as conditions in the claim.
* In the display following line 416, maybe good to mention that the Poissonization trick is used there
* In line 434, “When $d_{\chi^2}(p_{\mathcal{A}}, q_{\mathcal{A}}) \geq \varepsilon$” is written twice.
* In the display following line 446, should $\theta \cdot \eta$ instead be $\theta_i \cdot \eta$?Otherwise I don’t see how the equation can make sense ($\theta$ is a vector, $\eta$ is a scalar, and the output of the operation needs to be a scalar).

**Limitations:**

They have addressed their limitations.

---

> ### Author Rebuttal · Authors · 2024-08-07
>
> We thank the reviewer for their encouraging comments.
>
> We understand the confusion, and will rewrite the statement of the two lemmas in the revision to make it clear that these are minimax lower bounds. For Lemma 2.7, we will add a paragraph to provide some
> intuition, pointing out that it is based on the classical Le Cam's two-point method commonly used in the
> distribution testing literature, and briefly explaining ``why'' one would expect these to be hard instances. Indeed, instance-optimal bounds could be an
> interesting research direction to pursue.
>
> We will address the questions raised below:
>
> 1. $\log \left( \frac{16 k}{\varepsilon / \log (k / \varepsilon)} \right) =
> \log (16) + \log \left( \frac{k}{\varepsilon} \right) + \log \log (k /
> \varepsilon) \leqslant 2 \log (k / \varepsilon)$ for large enough $k$, as
> $\log \log x \ll \log x$.
>
> 2. You are correct. We could indeed obtain the same result by losing a little
> bit on the constant and dropping the assumption on $p (\bar{\mathcal{A}}) + q
> (\bar{\mathcal{A}})$.
>
> Minor comments:
>
> 1. It is referring to the paragraph from Line 103-106.
>
> 2. 3. 4. 5. 6. Yes, you are correct. We will make the corresponding changes in the final version.

---

> > ### Comment · Reviewer_BQBQ · 2024-08-12
> >
> > Thank you for your response. My questions have been clarified and I’ll maintain my current score.

---

### Author Rebuttal · Authors · 2024-08-07

We would like to thank all the reviewers for their detailed and thoughtful
comments (especially for taking a close look at the
proofs). We respond to each reviewer's comments individually below; and here focus on a common point raised across the reviews, that of the motivation or practical relevance of the question.

It is always delicate to argue about the importance of a newly formulated question, such as the one we are introducing and addressing in this paper. However, ``testing in entropy difference'' can be seen as a variant of the well-studied goodness-of-fit question, where the alternative hypothesis focuses on some particular premise (namely, that what one aims to detect as evidence is a discrepancy in entropy). This question actually originated from the main application we consider in the paper, that of testing Bayes networks, for which we provide an algorithm where testing in entropy difference is an important building block. Given the existing literature on estimating (Shannon) entropy and on testing graphical models, we are confident that this new problem will find more applications.

We also want to emphasize that the correctness issues raised by the reviewers stemmed from some lack of detailing on our end, but did not reflect a flaw in our arguments (see detailed individual comments). However, we feel compelled to mention that, in the process of improving the writing and presentation, we have since identified a potential issue in the informal description (ll. 142--148; we elaborate on this in the next paragraph). Fortunately, there is an easy fix which does not affect the end results: namely, instead of Hellinger distance, we can instead rely on an (similar-length) KL-divergence-based argument.

*Details*. The issue is that having all subsets $L \subset \\{ 0, 1 \\}^n$ with size of $d +
1$ close by $O (\varepsilon^2 / n)$ in Hellinger distance does not
guarantee that $d_H^2 (p_G, q_G) \leqslant O (\varepsilon^2)$ for every graph
$G$, as claimed in Line 142-148.
Instead, we can do local Kullback-Leibler tests on these subsets $L$, excluding small-probability elements: as it turns out out, this is enough, and does not require any change to the algorithm itself (only to its analysis).

If the reviewers feel this is useful, we would of course be happy to provide the full revised argument. We did so in the attached PDF; however, based on the guidelines, we understand that this PDF may only be read by the reviewers at their discretion.

---

### Decision · Program_Chairs · 2024-09-25

**Decision:**

Accept (poster)

**Comment:**

This manuscript proposes a new property testing problem called the entropy testing and obtains the tight sample complexity for this task. Surprisingly, the entropy testing can be used as a subroutine in testing Bayesian networks and lead to an improved sample complexity over [Canonne, Diakonikolas, Kane, and Stewart, 2020] and the removal of an unnecessary condition. The reviewers agree that the new problem is fundamental, the analysis is sound (originally one reviewer spotted an error, but the rebuttal clears the correctness concern), and the application to testing Bayesian networks is clever. The reviewer also raise some concerns, e.g. the overall algorithm is not very neat (needs to distinguish into cases) and hardly practical, which are not so important given the theoretical nature of this work. Overall I recommend acceptance.